# High-resolution landscape of an antibiotic binding site

Kevin B. Yang[1], Maria Cameranesi[1], Manjunath Gowder[1], Criseyda Martinez[1], Yosef Shamovsky[1], Vitaliy Epshtein[1], Zhitai Hao[1], Thao Nguyen[1], Eric Nirenstein[1], Ilya Shamovsky[1], Aviram Rasouly[1,2 ✉] & Evgeny Nudler[1,2 ✉]

Antibiotic binding sites are located in important domains of essential enzymes and have been extensively studied in the context of resistance mutations; however, their study is limited by positive selection. Using multiplex genome engineering[1] to overcome this constraint, we generate and characterize a collection of 760 single-residue mutants encompassing the entire rifampicin binding site of *Escherichia coli* RNA polymerase (RNAP). By genetically mapping drug–enzyme interactions, we identify an alpha helix where mutations considerably enhance or disrupt rifampicin binding. We find mutations in this region that prolong antibiotic binding, converting rifampicin from a bacteriostatic to bactericidal drug by inducing lethal DNA breaks. The latter are replication dependent, indicating that rifampicin kills by causing detrimental transcription–replication conflicts at promoters. We also identify additional binding site mutations that greatly increase the speed of RNAP. Fast RNAP depletes the cell of nucleotides, alters cell sensitivity to different antibiotics and provides a cold growth advantage. Finally, by mapping natural *rpoB* sequence diversity, we discover that functional rifampicin binding site mutations that alter RNAP properties or confer drug resistance occur frequently in nature.

In addition to its importance in combating clinical resistance[2–4], the study of antibiotic resistant mutants has also advanced our understanding of fundamental biological processes[5]. Because antibiotics function by targeting enzymes necessary for bacterial survival, resistance mutations have been instrumental in the characterization of essential cellular machinery such as RNA polymerase[6–10] (RNAP) and the ribosome[11–14]. Resistance mutations have been shown to inadvertently alter the function of antibiotic targets, providing new insights into the various functional[6,11,15], regulatory[16] and physiological attributes[12,17,18] of essential enzymes, identifying antibiotic binding sites as phenotypic hotspots.

The binding site of rifampicin (Rif), a specific inhibitor of bacterial RNAP, is well defined by high-resolution structures[8,10,19] that confirm earlier genetic findings from resistant mutants[20]. High-affinity binding of Rif prevents the extension of nascent RNA beyond a length of three nucleotides and Rif binding is lost in resistant mutants. Collectively, resistance mutants manifest a fascinating breadth of phenotypes that are a consequence of changes in the properties of the mutant enzyme at each step of the transcription cycle[15,16,21–24]. Although Rif is notorious for high-frequency resistance[4,25,26], the known spectrum of resistant mutants makes less than 5% of the non-redundant mutational space at this region of the β-subunit of RNAP. Therefore, our functional understanding of the Rif binding site is still incomplete.

Multiplex automated genome engineering (MAGE) in tandem with sequencing is free of the constraints of positive selection and low mutation rates[27–29]. This system enables the introduction and monitoring of mutations to a region of interest on the bacterial chromosome with extremely high efficiency and specificity[1,30,31] and has previously been used to characterize codon optimization and sequence diversity across the essential genes *infA* and *rpoD*[27,28].

In this work, we generate a collection of RNAP mutants that span all possible substitutions in each position of the Rif binding site. We provide a detailed genetic map of Rif binding interactions and identify an alpha helix spanning *rpoB* 521–526 where mutations can either increase or decrease Rif potency. In this region, we discover single-residue mutants that convert Rif from a bacteriostatic to a bactericidal drug by increasing its binding affinity and causing DNA damage. We show that the bactericidal effects are dependent on active replication, indicating that RNAP trapped by Rif at promoters causes detrimental transcription–replication conflicts. We delineate other single-residue mutations in the Rif binding site that produce fast RNAP and determine that high transcription speed depletes the cell of nucleotides, increasing susceptibility to antibiotics, but can also be beneficial as it provides a growth advantage at lower temperatures through poor termination efficiency. Despite high evolutionary conservation, we identify functional Rif binding site mutations that occur at a high frequency in nature. Our work demonstrates that the Rif binding site is a multifunctional domain of RNAP, and its characterization broadens our understanding of the antibacterial potential of Rif and the relationship between transcription speed and bacterial physiology.

[1]Department of Biochemistry and Molecular Pharmacology, New York University Grossman School of Medicine, New York, NY, USA. [2]Howard Hughes Medical Institute, New York University School of Medicine, New York, NY, USA. ✉e-mail: aviram.rasouly@nyulangone.org; evgeny.nudler@nyulangone.org

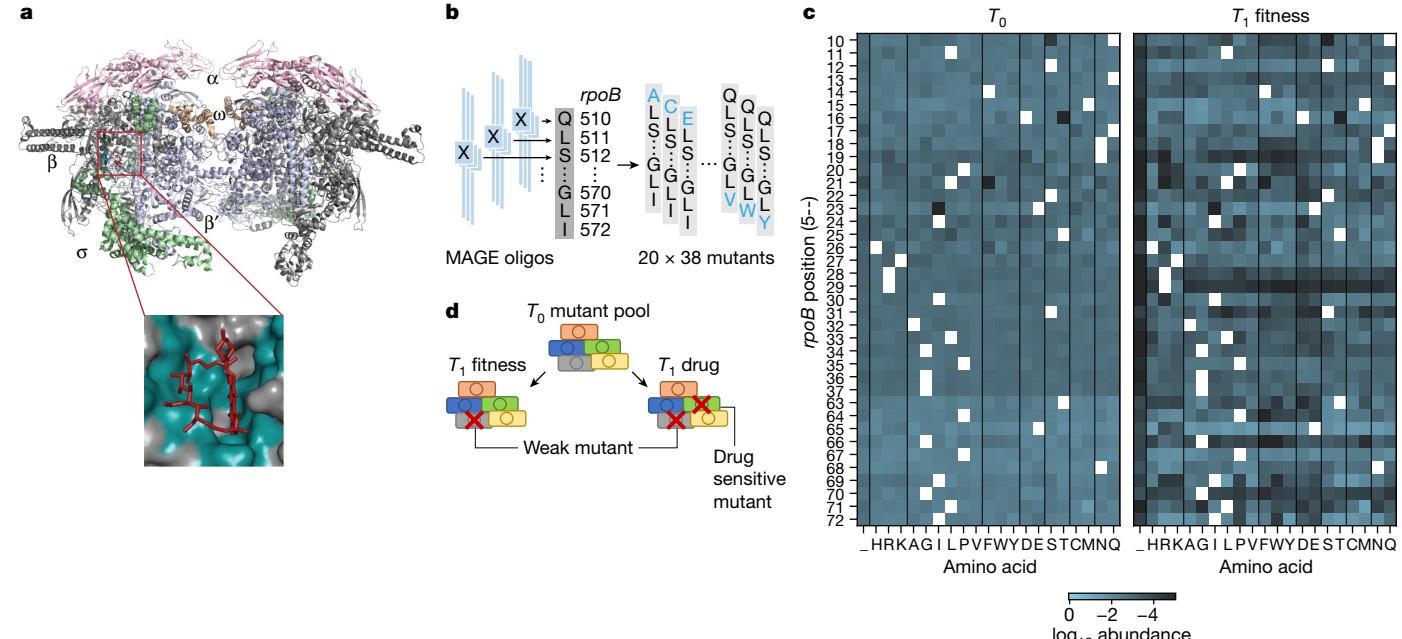

**Fig. 1 | Generation of all single codon substitutions in the Rif binding site.**
**a**, The Rif binding site within the *rpoB* subunit of *E. coli* RNAP (in blue), magnified in the lower panel (PDB 5UAC). **b**, Schematic diagram of the mutant pool generation. MAGE oligos were designed to generate a saturated pool of 760 single-codon mutants (20 substitutions × 38 positions) within the Rif binding site. **c**, Representative heatmap of mutant abundances generated with MAGE at timepoint 0 ($T_0$) and timepoint 1 ($T_1$), after fitness screening. For analysis of screening data, normalization was performed between $T_1$ and $T_0$ by calculating the $\log_2(T_1 \text{ fitness}/T_0)$ fold change. Rows correspond with *rpoB* positions 510–537 and 563–572; columns correspond to amino acids. White squares denote wild-type residues. **d**, Schematic of drug sensitivity and resistance screen. Mutants that fail to grow in LB are designated weak mutants whereas mutants that only fail to grow in drug are considered drug sensitive. Drug-specific effects can be isolated by normalizing '$T_1$ drug' mutant abundances to '$T_1$ fitness', that is, $\log_2(T_1 \text{ drug}/T_1 \text{ fitness})$.

## Deep mutagenesis of the Rif binding site

We generated and monitored a saturated pool of single amino acid substitutions in *rpoB* positions 510–537 and 563–572 using MAGE in tandem with sequencing[27] (Fig. 1a,b). These positions have been shown to interact with Rif either structurally or genetically[8,10] and encompass resistance mutants most commonly found in clinical isolates[9]. Following one cycle of MAGE, we prepared sequencing libraries to estimate the efficiency of oligo recombineering and retrieved most of the expected substitutions at a frequency between $10^{-3}$ and $10^{-5}$ (Fig. 1c and Extended Data Fig. 1a). We were able to reproducibly detect greater than 99% of generated mutants before selection, which is critical in our experimental design for the identification of sensitive mutants (Fig. 1d).

To distinguish drug sensitive from inviable mutants, we began by monitoring mutant fitness in regular growth conditions. We prepared samples from cells in liquid culture after two subsequent 100-fold dilutions, allowing cells to reach the same optical density (OD) before collection (Extended Data Fig. 1a). We also prepared samples after passaging the mutant pool on agar plates (Extended Data Figs. 1a and 2c,d). As expected, stop codons are quickly depleted (Fig. 1c and Extended Data Fig. 1a–d). We also observed that the distribution of fitness effects in this region of *rpoB* is bimodal, with a peak representing neutral substitutions at 0 and the other representing detrimental substitutions at −2 (Extended Data Fig. 2b). Other systematic mutagenesis studies in essential proteins have observed the same bimodal distribution[27,28]. When using 0 as the threshold for inviable mutants, 58% of substitutions are inviable, which is comparable to values of 53% and 48% for *rpoD* and *infA*, respectively[27,28].

Next, we screened our collection of *rpoB* mutants with Rif, bicyclomycin (BCM), and 5-fluorouracil (5FU) and normalized the data to isolate drug-specific effects (Fig. 1d and Extended Data Figs. 1–3). We selected BCM, an inhibitor of Rho termination, and 5FU, a nucleotide

analogue, because Rif resistant mutants have previously been shown to be deficient in Rho termination[32] and highly sensitive to nucleotide analogues[4]. Interestingly, this relatively small region of RNAP is rich in mutations that provide differential sensitivity towards these compounds (Fig. 2a and Extended Data Fig. 1). Rif resistance occurs in known positions; however, the range of substitutions conferring resistance is much wider (Figs. 2a and 3a). We note that some mutations, such as those at position 531, are relatively featureless despite being known to cause Rif resistance[10]. This is because these mutants are generally slow growing, as demonstrated by fitness screening, and therefore difficult to detect as they are highly resistant in the screening format (Extended Data Figs. 1a and 2a; Methods). Resistance to BCM is observed in mutations in the region around position 567 (Fig. 2a,b and Extended Data Fig. 3a,b,e,f). 5FU resistance also appears to occur in this region (Fig. 2a,b and Extended Data Fig. 3c,d,g,h).

By calculating the average fitness of all substitutions in a given position (Fig. 2b), we provide a measure for mutability of the wild-type residue. As expected for a highly conserved region of an essential protein, most positions are highly immutable with substitutions to R529, G566 and G570 producing the largest average fitness defects (Fig. 2b and Extended Data Fig. 2). On the other hand, the alpha helix at positions 523–527 appears to be highly mutable (Fig. 2b and Extended Data Fig. 2). Increased average positional fitness indicates that the original wild-type residue is disadvantageous in our growth conditions in some way.

In a similar manner, average positional resistance towards each drug measures the wild-type residue's importance in drug activity. Multiple mutations in positions 516, 526 and 572 increase average resistance towards Rif (Figs. 2b and 3a), indicating that the wild-type residue in those positions is important for Rif binding. High-level resistance in some positions is caused by a single particular substitution, such as R529K or P564K (Fig. 3a), implying that the substituted residue

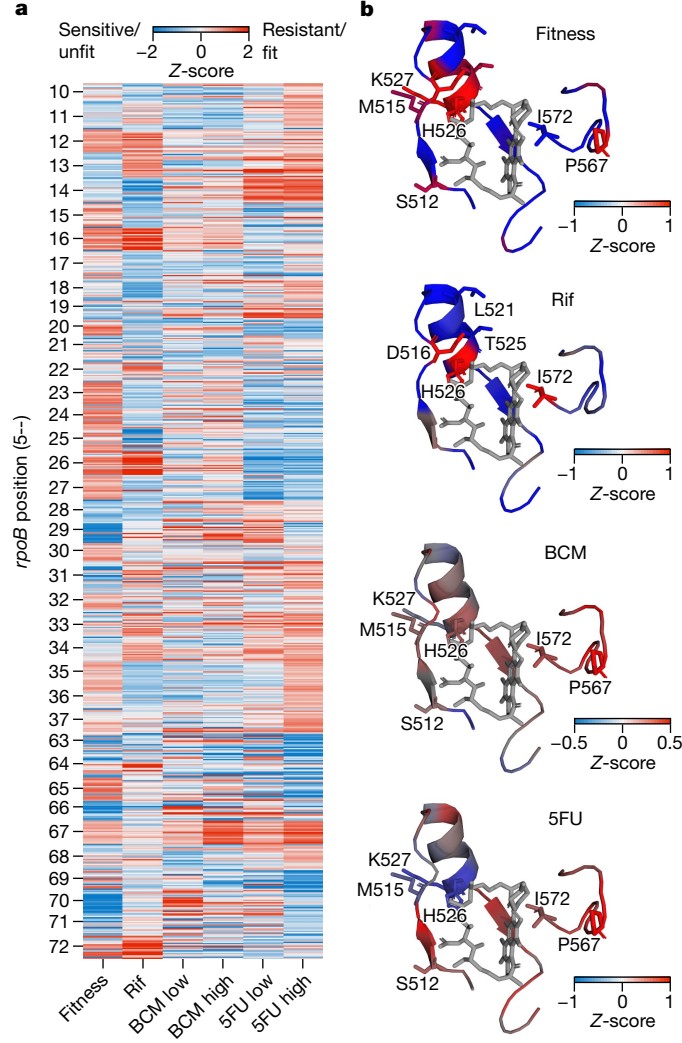

**Fig. 2 | The Rif binding site is abundant with mutations affecting bacterial physiology. a**, Aggregate heatmap of *rpoB* mutant fitness and sensitivity to Rif, BCM and 5FU. Columns correspond to each screening modality and rows represent each amino acid position. **b**, Protein structure of the Rif binding site with a colour map of average positional fitness or sensitivity to Rif, BCM or 5FU. Resistance or high fitness mutations are represented in red. *Z*-scores are calculated for each screening modality from MAGE-seq experiments conducted in *n* = 3 biologically independent replicates.

interferes with Rif binding. Mutations at position 567 increase average resistance towards both BCM and 5FU (Fig. 2b and Extended Data Fig. 3), indicating that the original proline increases sensitivity towards these drugs. Conversely, average positional sensitivity measures the wild-type residue's interference with drug activity. For example, the wild-type threonine at 525 appears to interfere with Rif binding as many mutations at this position increase sensitivity (Figs. 2b and 3a). Positional sensitivity or resistance provides an indication of the functional and structural importance of each position in single amino acid resolution.

## Rif hypersensitive binding site mutants

When examining the behaviour of individual mutations in response to Rif, we were intrigued to find Rif hypersensitive (Rif^HS) mutants in positions 514, 521 and 525 (Fig. 3a), with substitutions L521Y (21Y) and T525D (25D; throughout, mutants from our collection are abbreviated to the final two numbers of the position and the substituted amino acid) producing the strongest phenotype (Fig. 3b). These substitutions have

no fitness defect, indicating that their dilution when grown with Rif is not an artefact of mutant weakness (Fig. 3b and Extended Data Fig. 4a). They are also located on an alpha helix in close proximity to mutations that confer high-level resistance (Figs. 2b and 3a).

Rif^HS mutants 21Y and 25D were reconstructed in a wild-type MG1655 background for further characterization. In agreement with results from the primary screen, wild-type cells resumed growth 2 h after high exposure to Rif whereas Rif^HS mutants took more than 8 h to recover (Extended Data Fig. 4b). These mutations also result in a decreased minimal inhibitory concentration (MIC) for Rif (Fig. 3c and Extended Data Fig. 4c).

To directly test Rif's interaction with Rif^HS RNAP, we cloned mutant 21Y for recombinant expression and purification. In a modified runoff experiment in which polymerase is saturated with Rif, washed and then allowed to resume transcription, 21Y is slower to recover, indicating tighter Rif binding (Extended Data Fig. 4d). This is not due to differences in catalytic activity as the rate of 3-mer synthesis is unchanged, 3-mer being the maximum product length of a polymerase obstructed by Rif[8]. To test whether Rif causes Rif^HS RNAP to linger at promoters in vivo, we performed chromatin immunoprecipitation sequencing (ChIP–seq) on RNAP after treating cells with Rif for 1 h, washing and allowing cells to recover for an additional 1 h. After recovery, more Rif^HS RNAP can be found persisting at promoters when compared to wild-type cells (Fig. 3d and Extended Data Fig. 4e).

Rif is known to be bacteriostatic in *Escherichia coli*. To test Rif^HS mutants for cell killing, cells were treated with Rif, washed and serial dilutions were assayed for growth on either agar plates or liquid broth (LB)[18]. Notably, although Rif remains bacteriostatic in wild-type cells, it becomes bactericidal for both Rif^HS mutants in *E. coli*, decreasing viability by at least one order of magnitude (Fig. 3e and Extended Data Fig. 5a–d). This effect is specific to Rif as chloramphenicol and tetracycline remain bacteriostatic (Extended Data Fig. 5e).

## DNA damage underlies Rif lethality

Previous studies report increased sensitivity of DNA-damage repair mutants in response to Rif treatment[33]. To investigate the bactericidal effects of Rif, we used terminal deoxynucleotidyl transferase deoxyuridine-triphosphate (dUTP) nick-end labelling (TUNEL)[34] to assess DNA fragmentation after 1 h of Rif treatment. Wild-type cells exhibit weak TUNEL staining of around 15% of positive cells after Rif treatment, which is lower than any bactericidal antibiotic tested in previous work[35] (Fig. 3f). Both Rif^HS mutants show increases in TUNEL staining with Rif treatment of up to 35%, similar to other bactericidal drugs such as 5FU and norfloxacin (approximately 30%)[35].

To determine whether DNA breaks contribute to the bactericidal effects of Rif, we generated knockouts of major double strand break (DSB) repair enzymes *recA* or *recBCD* in Rif^HS mutants. Notably, Rif becomes potently bactericidal in both Rif^HS mutants in the absence of DSB repair (Fig. 3g). In addition, wild-type cells lacking *recA* become sensitive to Rif killing (Fig. 3g), providing additional evidence that Rif causes low levels of DNA damage in wild-type cells (Fig. 3f). We note that these effects are Rif specific, as Rif^HS mutants deficient in DSB repair remain insensitive to other bacteriostatic drugs (Extended Data Fig. 5f).

As a secondary indicator of DNA damage, we measured the expression of key genes in the SOS response. On Rif treatment, we see a significant increase in the expression of genes involved in DSB repair (Extended Data Fig. 6a). We also measured protein levels of the SOS gene repressor LexA following Rif treatment and observed that LexA protein levels significantly decrease following Rif treatment, indicating that it is cleaved in response to DNA damage (Extended Data Fig. 6b,c). In agreement with previous work, the bactericidal effect of Rif does not entail increased cellular respiration and the accumulation of toxic reactive oxygen species (ROS)[36,37] (Extended Data Fig. 6d–f).

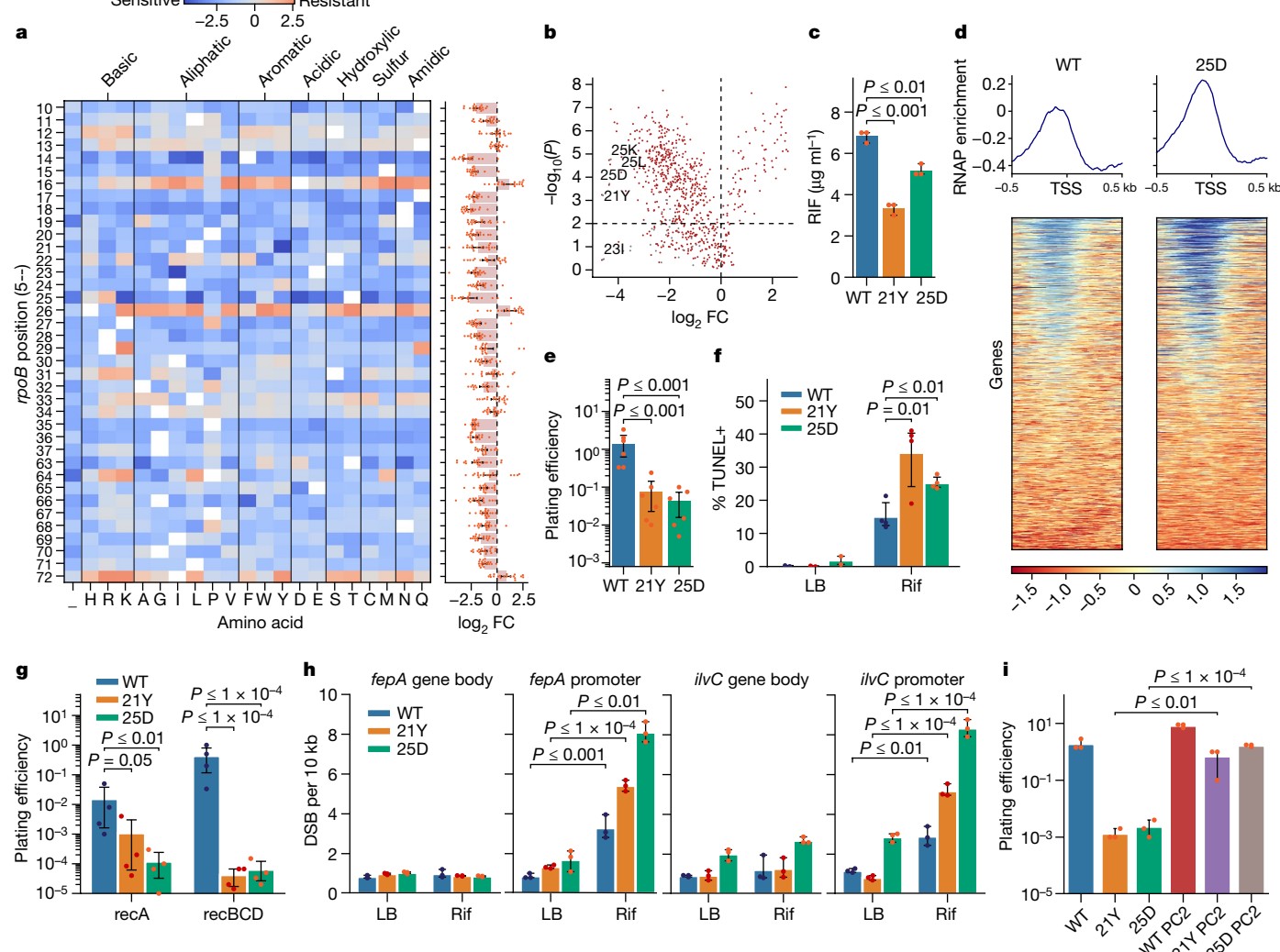

**Fig. 3 | Bactericidal effect of Rif on a new class of hypersensitive mutants.**
**a**, Sensitivity and resistance landscape of *rpoB* mutants towards Rif. Rows correspond with *rpoB* position and columns denote all 20 amino acid substitutions, grouped by property. White squares denote wild-type residues. Bar plot of average positional fold change is provided to the right of the heatmap.
**b**, Volcano plot of data in **a**. Stop codons are in grey. **c**, MIC of Rif in hypersensitive mutants ($P = 1.2 \times 10^{-4}$, $2.1 \times 10^{-3}$ from left to right). **d**, Meta-analysis of ChIP–seq signal from wild-type (WT) and *rpoB* T525D RNAP around transcription start sites (TSS) of highly transcribed genes after 1 h recovery from 1 h treatment with 80 µg ml⁻¹ Rif. **e**, Plating efficiency of hypersensitive mutants following 1 h treatment of 80 µg ml⁻¹ Rif ($n = 6$; $P = 5.7 \times 10^{-4}$, $1.8 \times 10^{-4}$). **f**, DNA damage measured by TUNEL staining in hypersensitive mutants following 1 h treatment with 80 µg ml⁻¹ Rif. TUNEL positive cells are the percentage of cells exceeding

signal detected in more than 99% of untreated cells ($n = 4$; $P = 1.3 \times 10^{-2}$, $4.7 \times 10^{-3}$). **g**, Plating efficiency of hypersensitive mutants in *recA* or *recBCD* knockout strains after 1 h treatment of 80 µg ml⁻¹ Rif ($n = 4$, $P = 5.3 \times 10^{-2}$, $7.3 \times 10^{-3}$, $4.2 \times 10^{-5}$, $5.7 \times 10^{-5}$). **h**, Number of DSBs per 10 kilobases (kb) at promoters and over the gene body measured using SLR-qPCR following 1 h treatment with 80 µg ml⁻¹ Rif ($P = 2.4 \times 10^{-3}$, $1.7 \times 10^{-5}$, $1.2 \times 10^{-4}$, $3.7 \times 10^{-3}$, $2.4 \times 10^{-5}$, $4.2 \times 10^{-5}$). **i**, Plating efficiency of hypersensitive mutants in the PC2 background containing a temperature-sensitive allele of the replication protein dnaC ($P = 1.8 \times 10^{-3}$, $7.6 \times 10^{-5}$). Replication was blocked by shifting cells to 42 °C for 1.5 h before 30 min treatment with 80 µg ml⁻¹ Rif. Bar plots represent the mean of $n = 3$ biologically independent experiments unless noted otherwise, with error bars denoting 95% confidence intervals. *P* values are calculated using the independent *t*-test (two-sided).

To investigate the location of DNA damage in Rif^HS mutants, we used semi-long-range quantitative PCR (SLR-qPCR) to detect DNA breaks at specific genomic loci (Extended Data Fig. 7a)[38]. We arbitrarily selected two highly expressed σ⁷⁰-dependent genes at distant chromosomal loci and compared the frequency of DSBs at cognate promoters to gene bodies (Extended Data Fig. 7b). Notably, we observed a significant increase in DSBs at promoters after Rif treatment in both wild type and Rif^HS mutants, whereas no increase occurred over the respective gene body (Fig. 3h). Furthermore, Rif^HS mutants exhibited a greater level of promoter-specific DSBs than wild type (Fig. 3h).

Previous work has shown that the transcription initiation complex can collide with or interrupt the replisome, resulting in interference with replication fork progression or deleterious mutations[39,40]. To determine

whether replication conflicts at promoters are involved in DNA damage by Rif, we introduced dCas9 targeting the oriC to Rif^HS strains to partially block replication before Rif treatment[41] (Extended Data Fig. 7c). We found that blocking replication significantly decreases the frequency of DSBs at promoters (Extended Data Fig. 7d,e). Next, we constructed Rif^HS mutants in cells containing a temperature-sensitive allele of the replication protein dnaC2 (PC2)[39,42]. We verified that shifting cells to the non-permissive temperature 42 °C inhibits replication (Extended Data Fig. 7f). We then compared the bactericidal effects of Rif with and without replication. Notably, we observe that Rif is no longer bactericidal in Rif^HS mutants when replication is inhibited (Fig. 3i). Rif^HS mutants in the PC2 background remain hypersensitive at the permissive temperature of 30 °C (Extended Data Fig. 7g).

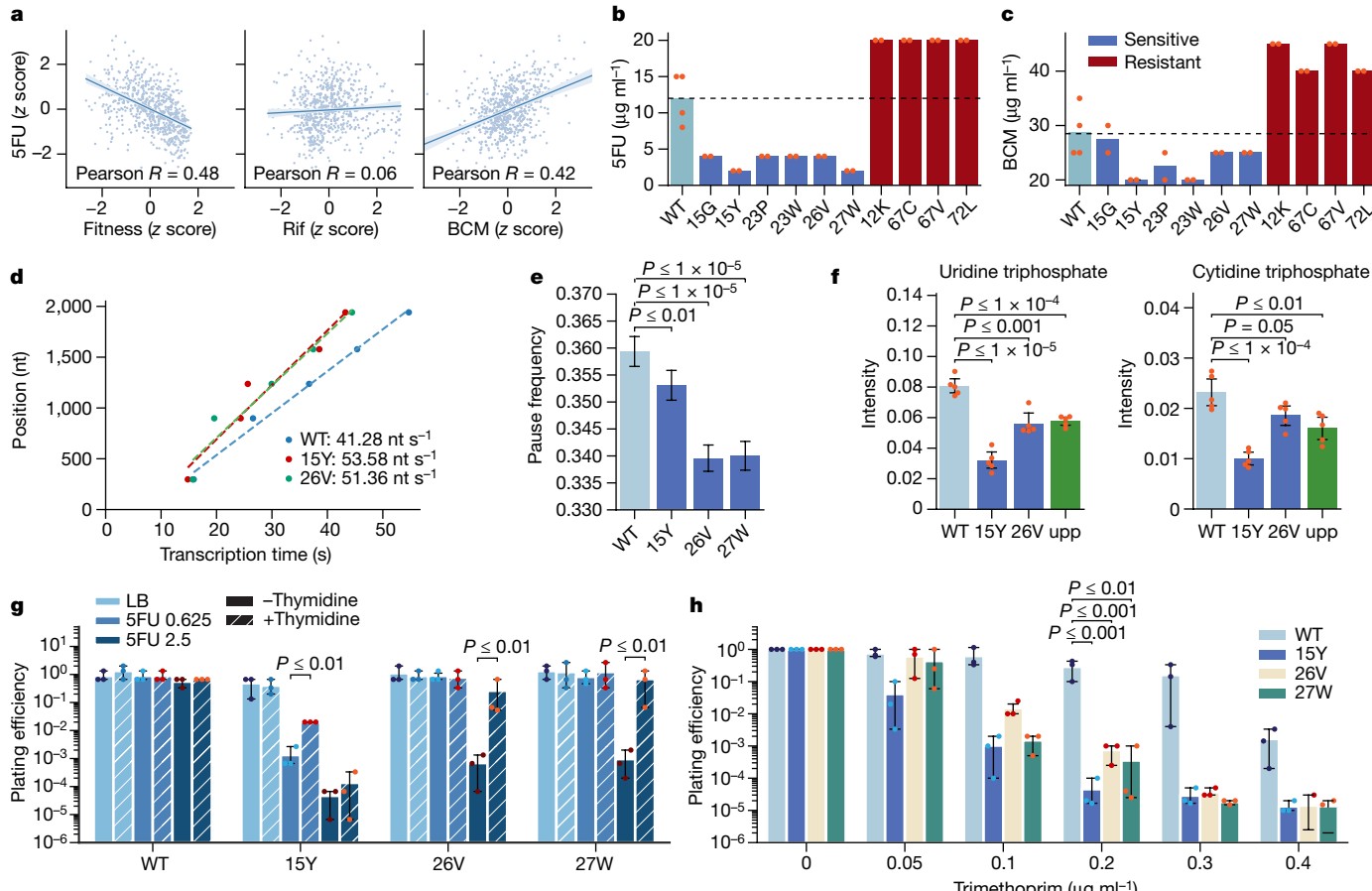

**Fig. 4 | Fast RNAP mutants deplete nucleotides and alter drug sensitivity.** **a**, Scatterplots of *rpoB* mutant response to 5FU compared to fitness or mutant response to Rif and BCM. Linear regression lines, 95% confidence intervals and Pearson *R* values are overlaid. Two-sided *P* values for 5FU correlations with fitness, Rif and BCM datasets are $2.4 \times 10^{-43}$, 0.11 and $5.2 \times 10^{-32}$, respectively. **b,c**, MIC of 5FU (**b**) and BCM (**c**) for sensitive or resistant strains reconstructed in MG1655 (*n* = 2). **d**, Transcription elongation speed of wild type and mutants. Transcription of five regions on the *lacZ* gene is measured after induction and plotted as a function of time and distance. The slope of the best-fit line represents transcription elongation speed. **e**, Mean pause frequency in wild type and 5FU sensitive mutants as measured by NETseq. Values denote the aggregated mean pause frequency per kilobase for the top 3,500 expressed genes. *n* = 2

biologically independent experiments for each condition ($P = 1.6 \times 10^{-3}$, $9.8 \times 10^{-25}$, $7.2 \times 10^{-22}$ from left to right). **f**, Pyrimidines in 5FU sensitive mutants as detected by LC–MS, normalized to labelled UTP (*n* = 5; $P = 2.4 \times 10^{-6}$, $5.6 \times 10^{-4}$, $8.9 \times 10^{-5}$, $6.4 \times 10^{-5}$, $5.0 \times 10^{-2}$, $8.4 \times 10^{-3}$). **g**, Plating efficiency of 5FU sensitive mutants with and without thymidine supplementation. Cells from overnight cultures were serially diluted and plated on LB agar plates with the indicated concentrations of 5FU in µg ml⁻¹ with and without 1 mg ml⁻¹ thymidine ($P = 3.1 \times 10^{-3}$, $8.6 \times 10^{-3}$, $4.7 \times 10^{-3}$). **h**, Plating efficiency of 5FU sensitive mutants with the indicated concentrations of trimethoprim ($P = 2.5 \times 10^{-4}$, $7.6 \times 10^{-4}$, $3.3 \times 10^{-3}$). Bar plots represent the mean of *n* = 3 biologically independent experiments unless noted otherwise, with error bars denoting 95% confidence intervals. *P* values are calculated using the independent *t*-test (two-sided).

In summary, we demonstrate that: (1) Rif^HS RNAP with Rif are more persistently bound at promoters; (2) the bactericidal effect of Rif in Rif^HS mutants is associated with double-stranded DNA breaks at promoters; (3) Rif induces the SOS response; and (4) Rif killing is much more severe in cells lacking DSB repair. Interestingly, Rif killing is not associated with a buildup of ROS, indicating that it acts in a manner distinct from other bactericidal antibiotics. Instead, we observe that bactericidal effects and DNA damage from Rif are replication dependent, implying that these effects are a result of transcription–replication conflicts at promoters.

## Fast mutants of the Rif binding site

The potential of mutations in the Rif binding pocket to affect Rif binding is obvious, but the mechanism by which they cause differences in sensitivity towards other drugs is not readily apparent. By pairwise comparison of the phenotypic screens, we noticed correlations between fitness, BCM and 5FU sensitivity whereas no such correlations exist with Rif (Fig. 4a). High fitness mutants are extremely sensitive to 5FU, especially

substitutions disrupting the α-helix at positions 523–527 (Fig. 2a). In addition, *rpoB* mutant sensitivity towards BCM and 5FU is positively correlated, hinting at a common underlying mechanism (Fig. 4a). We selected mutants with strong phenotypes from each screening modality for reconstruction and further characterization in MG1655 (Extended Data Figs. 1–3). We found the MIC of all reconstructed *rpoB* mutants to closely reflect screening results. Mutants 15G, 15Y, 23P, 23W, 26V and 27W are highly sensitive to 5FU, with around a fivefold decrease in MIC (Fig. 4b); these mutants are also sensitive to BCM (Fig. 4c). Mutants 12K, 67C, 67V and 72L all display high resistance towards both BCM and 5FU (Fig. 4b,c).

BCM acts by inhibiting the Rho transcription termination factor. Previous work has shown that polymerase speed determines the efficiency of Rho termination[32]. To test whether mutants sensitive to both 5FU and BCM have faster RNAP, we measured their elongation rate using a multiprobe qPCR assay along the *lac* operon and their pause frequency using native elongating transcript sequencing (NETseq)[4,43]. Indeed, these mutants have a higher elongation rate (Fig. 4d) and pause less frequently (Fig. 4e) than wild-type RNAP.

We considered the possibility that high transcription rates in fast RNAP mutants deplete nucleotide pools, resulting in 5FU sensitivity. A known target of 5FU is *thyA*, encoding the enzyme responsible of dTMP synthesis. dUMP, the substrate for this enzymatic reaction, is produced from pyrimidine ribonucleotides. Therefore, a strain on the pool of pyrimidine ribonucleotides would sensitize cells to 5FU[44,45]. Using hybrid liquid chromatography–mass spectrometry (LC–MS), we found that 5FU sensitive mutants are depleted of the two detectable pyrimidines, uridine and cytidine triphosphate, to the level of an *upp* mutant deficient in uracil salvage (Fig. 4f). 15Y, a mutant incredibly sensitive to 5FU, is depleted of purines as well (Extended Data Fig. 9d). Indeed, thymidine supplementation rescued 5FU sensitivity of mutants 15Y, 26V and 27W (Fig. 4g), as well as the growth defect of 15Y under normal growth conditions at 37 °C (Extended Data Fig. 9e). Of note, modest changes in the expression level of nucleotide biosynthesis genes do not account for 5FU sensitivity of fast mutants (Extended Data Figs. 8 and 9a–c).

Because 5FU inhibits bacterial growth through a complex mechanism that includes not only dTMP depletion, but also misincorporation to DNA and RNA, we sought to establish sensitivity to dTMP depletion using thymidine-specific assays. First, we tested whether fast mutants were more susceptible to 'thymineless death' by constructing fast mutants in a genetic background of *thyA*, which requires thymidine supplementation for growth. When thymidine is removed, cells are able to subsist on the existing reservoirs of dTMP during a 'resistance phase' before viability is lost[46]. In support of faster depletion of intracellular thymidine, fast mutants in a *thyA* background have a shorter resistance phase than wild-type RNAP (Extended Data Fig. 9f). Notably, the well-studied *rpoB* 513P, a slow RNAP[15,47], is more resistant to 5FU and has a protracted resistance phase in the thymineless death assay (Extended Data Fig. 9g,h). Second, we tested sensitivity of fast mutants towards trimethoprim. This clinically important antibiotic inhibits dihydrofolate reductase, which is required for de novo synthesis of thymidine in addition to certain amino acids[48]. Because LB medium is limited for thymidine, its availability affects overall trimethoprim sensitivity. Consistent with our model for nucleotide depletion, fast RNAP mutants are also more sensitive to trimethoprim (Fig. 4h). Together, these results indicate that RNAP speed determines the availability of nucleotides within the cell.

All fast RNAP mutants we isolated appear to have a fitness advantage in the original screening format (Extended Data Figs. 1a and 2), which was done at lower temperatures suitable for MAGE recombineering strains[1]. When reconstructed in MG1655, fast mutants have no such growth advantage at 37 °C, but all grow faster and saturate at a higher OD than wild-type cells at a lower temperature of 25 °C (Extended Data Fig. 10a). 5FU[R] mutants grow slowly at 25 °C (Extended Data Fig. 10b). At 42 °C, fast mutants 15Y and 27W are temperature sensitive and grow more slowly (Extended Data Fig. 10b). Previous work has shown temperature sensitivity of naturally occurring Rif resistant mutant 526Y, which is also known to be a fast RNAP[7]. Because fast polymerases terminate inefficiently, we hypothesized that increased readthrough might provide a growth advantage at 25 °C. Indeed, wild-type cells with subinhibitory amounts of BCM grow faster at 25 °C (Extended Data Fig. 10c). From these data, we identify that fast RNAP provides a growth advantage at lower temperatures, at least in part through poor termination efficiency.

In summary, our characterization of fast RNAP mutants of the Rif binding site led to the observation that the speed of transcription controls the availability of nucleotides within the cell. Nucleotide pools are limiting and marginal increases in RNAP speed greatly sensitize cells to antibiotics that target nucleotide biosynthesis. In addition, fast RNAP also affects desirable growth environments for bacteria, raising the question of whether such mutations appear in nature.

## Rif binding site mutations in nature

We aimed to inform our study of *rpoB* mutants by exploring natural sequence diversity. Following alignment of around 4,000 orthologues of *E. coli rpoB* mined from KEGG[49], we observed high evolutionary conservation at the Rif binding site by Jensen–Shannon divergence[50] (Fig. 5a). We next grouped species by phylum or class and determined the number and type of substitutions that separate orthologues from *E. coli* MG1655 *rpoB* within the Rif binding site. Within this 38 amino acid window, most species available for analysis contained fewer than five substitutions, with mild substitutions appearing in closely related proteobacteria (Fig. 5b,c). As one would expect in a functionally conserved region of RNAP, the most common substitutions are between amino acids of the same functional group with the exception of S522A, a hydroxylic to aliphatic substitution found in close to 10% of species surveyed (Fig. 5b,c and Extended Data Fig. 2).

When comparing average positional fitness to evolutionary conservation, we see a negative correlation as expected (Extended Data Fig. 10d). Generally, highly conserved positions are less mutable, except in some instances. For example, histidine at 526 is highly conserved but is mutable in our fitness screen. Mutating this position is probably deleterious in other ways, which is why it is rarely found in evolution. For example, mutations in this position are highly sensitive to 5FU (Extended Data Fig. 10d).

Average positional fitness is limited in determining the effects of single-residue variants found in nature. To speculate how natural variance in the Rif binding site might evolve, we compared the frequency of natural variance to substitution fitness in each screening modality (Fig. 5d). Among the most common natural variants with strong phenotypes in our screen, 24L, 26N and 67Q are the most notable. 24L is found in more than 10% of species surveyed, has characteristics of a fast RNAP and appears infrequently in Proteobacteria, but more commonly in Actinobacteria and Firmicutes (Fig. 5c,d and Extended Data Figs. 2 and 3). 26N is found in 1% of species surveyed, has characteristics of a fast RNAP with high-level Rif resistance and appears in Tenericutes (Fig. 5c,d and Extended Data Figs. 2 and 3); notable genera include *Mycoplasma*, which have been shown to be Rif resistant[9]. 67Q provides resistance to both BCM and 5FU and is found in -Epsilonproteobacteria and the closely related Aquificae; notable species include *Helicobacter* and *Campylobacter* (Fig. 5c,d). Some strains of Epsilonproteobacteria have been shown to be resistant to 5FU[51]. Future work will need to be done to characterize the effects of the identified mutations in these organisms.

## Discussion

To build upon decades of work with RNAP resistant mutants while avoiding the bias of characterizing antibiotic binding sites through the lens of resistance, we present a complete collection of single-residue mutants of the Rif binding site. With our collection, we observe that the Rif binding site is surprisingly mutable and that functional mutations are not exclusive to Rif resistance and cannot be fully appreciated without saturating mutagenesis (Figs. 1 and 2 and Extended Data Figs. 1–3). We expand upon the known Rif 'resistome' by identifying that nearly every substitution at positions 516, 526 and 572 generates high-level resistance (Figs. 2 and 3a). Many of these resistant mutations do not occur naturally, probably because they require multiple nucleotide substitutions. Moreover, we identify a single alpha helix that has a critical role in Rif binding, where mutations either cause high-level resistance (S522 and H526) or increased sensitivity (L521 and T525) (Figs. 2 and 3a). These data provide a detailed genetic mapping of ligand–enzyme interactions that will be of value to medicinal chemists working to increase antibiotic potency.

We discover single-residue mutations of RNAP that convert Rif into a bactericidal drug in *E. coli* (Fig. 3). We show that these mutations act by prolonging Rif binding as the mutant polymerase is slow to recover from Rif treatment both in vivo and in vitro (Fig. 3d and Extended Data Fig. 4d), the behaviour of other bacteriostatic drugs is unchanged (Extended Data Fig. 5e,f), the mutants have no fitness defect (Extended Data Fig. 2) and we observe no gene expression reprogramming (Extended

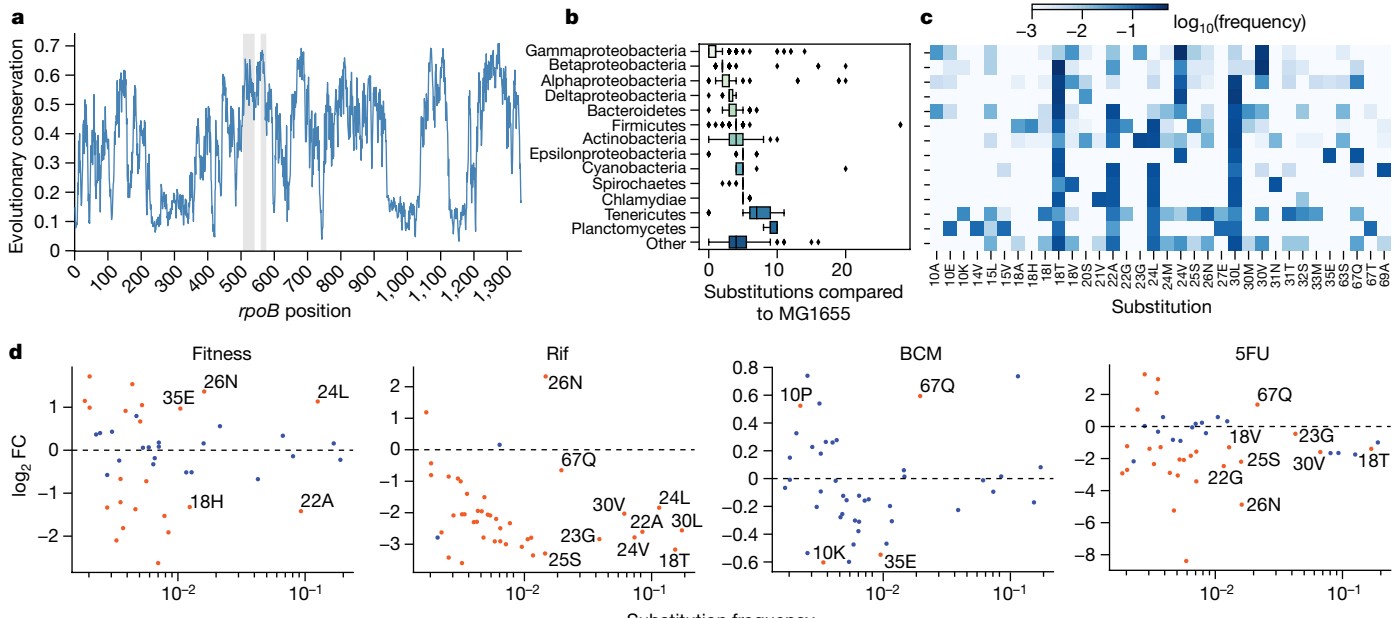

**Fig. 5 | Rifampicin binding site mutations in nature. a**, Evolutionary conservation of *rpoB* by Jensen–Shannon divergence based on alignment of 4,055 sequences. Higher values indicate higher conservation. Positions are based on MG1655 *rpoB* and the Rif binding site is highlighted in grey. **b,c**, Total number of substitutions (**b**) and frequency of specific residue substitutions (**c**) that separate the surveyed 4,055 *rpoB* orthologues from MG1655 grouped by phylum. Proteobacteria are further subdivided by class. Phyla with fewer than 40 representative species are combined in 'other'. Substitutions found in fewer than 20 species are omitted for brevity. In the box plot, the lower and upper ends of the box denote Q1 and Q3, respectively. The whiskers span 1.5 × (Q3 − Q1) from each side of the box. **d**, Scatterplots of substitution frequency in nature compared to substitution fitness, sensitivity and resistance. Substitutions found in less than 20 species are omitted for brevity. Orange denotes substitutions with significant (*P* < 01) phenotypes in each screen. *P* values are calculated using the independent *t*-test (two-sided).

Data Figs. 8 and 9a–c). In a similar way, next-generation macrolides are bactericidal by virtue of higher affinity to the ribosome than that of the parental bacteriostatic drug[14,52].

DNA-damage repair mutants have previously been observed to have an increased sensitivity to Rif[33]. In this work, we directly observe that Rif causes double-stranded DNA breaks at promoters, induces the SOS response and kills Rif[HS] mutants (Fig. 3 and Extended Data Figs. 5a–d and 6a–c). We show that bactericidal effects are exacerbated in Rif[HS] mutants deficient in DSB repair and also in wild-type *recA* cells, indicating that Rif damages DNA in wild-type cells as well, but not to the extent of causing cell death (Fig. 3g). In agreement with previous work, we find that the bactericidal effects of Rif are not dependent on ROS (Extended Data Fig. 6d–f), indicating a mechanism distinct from other bactericidal antibiotics[36,37]. Instead, we observe that Rif causes DNA damage and kills only when cells are actively replicating (Fig. 3i and Extended Data Fig. 7). RNAP arrested at the promoter is known to present an obstacle for the replisome and transcription–replication conflicts have been shown to cause detrimental mutations[39,40]. Our results indicate that Rif damages DNA through a similar mechanism. Future work will need to be done to test this model in other bacteria. These findings imply that improving Rif binding affinity can shorten treatment times and extend its antimicrobial spectrum of activity. Moreover, the discovery that Rif acts through a unique mechanism to cause DNA damage may inform the design of combination antimicrobial therapy.

We discover in our collection other single-residue mutations that yield fast RNAP, similar to the well-characterized Rif resistant H526Y mutant and other mutants of RNAP[15,21,53]. We observe that high transcription rates deplete pyrimidine nucleotides, such that these mutants become particularly vulnerable to further depletion through 5FU inhibition of *thyA*[54] (Fig. 4). Importantly, growth media supplemented with thymidine are able to rescue 5FU sensitivity of fast RNAP (Fig. 4g). We note that thymidine supplementation does not render cells fully resistant to 5FU. Misincorporation of dUTP and its 5-fluoro analogue

during replication, as well as 5FU misincorporation during transcription, may contribute to the antimicrobial effects of 5FU. Nucleotide depletion by fast RNAP also makes cells more susceptible to thymineless death (Extended Data Fig. 9f) and to the clinically important antibiotic trimethoprim (Fig. 4h). High-speed transcription also sensitizes cells to the antibiotic BCM because of poor termination efficiency; however, we discover that poor termination provides a growth advantage at lower temperatures (Extended Data Fig. 10c). Together, these results demonstrate that transcription speed determines nucleotide availability, can alter antibiotic sensitivity and can change the optimal growth temperature of bacteria.

Despite high evolutionary conservation of the Rif binding site, we identify functional RNAP mutations that exist at a high frequency in nature. 524L and 526N both share characteristics with fast RNAP mutants and are located on the alpha helix we identified to be important for Rif binding (Figs. 2 and 3). As such, 526N provides high-level Rif resistance and increases sensitivity to 5FU, but not to the same degree as 526Y (Extended Data Fig. 3). It is intriguing to see that clinical resistance, which develops relatively quickly, provides a larger fitness cost than resistance that developed over millions of years of evolution. It is also interesting to speculate why Tenericutes and other phyla would carry this mutation. Perhaps they lived in close vicinity to Rif producing bacteria. *rpoB* 567Q, found in 2% of species surveyed, provides high-level resistance to both 5FU and BCM. We notice that species carrying these mutations live in harsh environments, such as hydrothermal vents and animal digestive tracts. These mutations were not a focus of this Article but they are probably slow, stringent mutants of RNAP, similar to 513P[16] (Extended Data Fig. 9g,h), and we speculate that they increase tolerance to living in nucleotide- or nutrient-limiting conditions.

Together, our high-resolution characterization of the Rif binding site reveals several RNAP subtypes, some of which can be found in nature. We find RNAP that is exquisitely sensitive to Rif, converting it to a bactericidal antibiotic in a gram-negative bacterium through

a unique mechanism of DNA damage. We also find that transcription speed determines the availability of nucleotides within the cell and can alter antibiotic susceptibility. Our study illuminates the antibiotic binding site as a multifunctional domain of essential enzymes and inspires similar characterization of other major antibiotic targets to broaden our understanding of antibacterial mechanisms, bacterial physiology and evolution.

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

## Methods

### Saturation mutagenesis screening approach

MAGE with the *E. coli* EcNR2 strain[1] was used to generate a pool of single-residue Rif binding site substitutions comprising *rpoB* 510–537 and 563–572. 760 MAGE oligos (Integrated DNA Technologies) were designed to generate 20 possible single-residue mutants in 38 positions. To limit the detection of background mutations and further increase the specificity of our approach, we specifically designed our MAGE library to produce single codon substitutions that include multiple nucleotide alterations when possible. A single round of MAGE mutagenesis was done in four bins comprising 510–518, 519–527, 528–537 and 563–572. Cells were recovered overnight and stocked in glycerol.

To begin a screening experiment, cells were thawed, bins were combined and allowed to recover to an OD at 600 nm (OD600) of 0.4 at which point an aliquot was taken as $T_0$. Cells were then subject to a 'recovery format' for Rif treatment or a 'MIC format' for BCM or 5FU (Sigma) treatment. In the 'recovery format', cells were treated with 50 μg ml$^{-1}$ Rif (Sigma) for 1 h, washed with LB, diluted tenfold and allowed to recover to the OD before dilution before being collected ($T_1$ drug). In the 'MIC format' cells were diluted tenfold and treated with BCM (15 or 30 μg ml$^{-1}$) or 5FU (1 or 5 μg ml$^{-1}$) and allowed to recover to the OD before dilution before being collected ($T_1$ drug). Untreated cells were grown in parallel to measure and control for mutant fitness ($T_1$ fitness). Untreated cells were also passaged on agar plates by plating cells and scraping colonies after overnight incubation. All screening was done in triplicate. The screening format we used depended on the selected drug. Because Rif is a bacteriostatic drug[37], we were not expecting sensitive mutants to decrease in abundance. Fortunately, previous work has shown that Rif produces a 'post-antibiotic effect' in which cells show a recovery delay after Rif is washed away and cells are allowed to resume growth[55,56]. For this reason, we designed a recovery format in which mutant *rpoB* pools were first treated with Rif, washed, diluted tenfold and then allowed to recover. Sensitive mutants would be expected to recover more slowly and would then be found at a lower abundance. As both BCM and 5FU are bactericidal[37,54,57], we expected sensitive mutants to die and decrease in abundance. Therefore, two concentrations were selected for each drug: one concentration at the MIC at which only resistant mutants would survive and one at a lower concentration at which sensitive mutants would be lost. We isolated drug-specific effects by normalizing mutant abundances between drug treated ($T_1$ drug) and untreated ($T_1$ LB).

### Screening library construction and sequencing

Genomic DNA was prepared from collected cells with the Monarch Genomic DNA purification kit (NEB). Libraries were prepared in two bins comprising *rpoB* 510–537 and 563–572. Here 100 ng DNA was first subject to two cycles of PCR in a 50 μl reaction containing Q5 polymerase (NEB) and 1 μM primers (Supplementary Table 2) designed to attach a unique barcode of 14 degenerate nucleotides to correct for PCR duplicates during analysis. Following barcode attachment, reactions were treated with 1 μl of exonuclease I (NEB) to remove unused barcodes for 1 h at 37 °C and then DNA was purified using a 1.5× ratio of PCRClean DX beads (Aline Biosciences). Sample barcode indexes and sequencing adaptor sequences were then attached with 18 cycles of exponential PCR before 2 rounds of purification with a 1× ratio of beads. Libraries were sequenced with the Novaseq 6000 SP300 cartridge (Illumina), aiming to collect 40 million reads per library and 2 × 10$^6$ unique barcodes for each experimental sample, allowing observation of 20–20,000 instances of each mutant.

### Screening data analysis

Sequencing adaptors were trimmed using CutAdapt[58]. Next, sequences were filtered for correct length and primer sequences, and then grouped into 'families' by the aforementioned nucleotide barcodes. Families representing an original genomic molecule were checked for 80% consensus by nucleotide, defaulting to the wild-type sequence in the case of no consensus and then translated to the designed amino acid sequence. Families were grouped by mutant and the number of families corresponding to a single mutant was divided by the total number of families to yield mutant frequency. Only single-residue mutants were considered for analysis. To determine mutant fitness, mutant abundances after growth ($T_1$ fitness) were normalized to abundances before growth ($T_0$). To determine drug-specific effects, mutant abundances after drug treatment ($T_1$ drug) were normalized to abundances after growth in LB ($T_1$ fitness).

### Strain construction

For all validation experiments, wild-type *E. coli* MG1655 was used as the parental strain (Supplementary Table 1). To generate selected *rpoB* mutants for reconstruction in MG1655, we used MAGE to introduce a kanamycin cassette at the 3′ end of *rpoC* in the *E. coli* EcNR2 strain. Selected *rpoB* mutants were generated in the resulting *E. coli* EcNR2 rpoc:kn. P1 phage-mediated transduction was used to move mutants to MG1655. For knockout strains, pKD46 was used as previously described[59].

### MIC and MBC assays

Overnight cultures of the indicated strains grown in LB were diluted 10,000-fold and growth in 0.2 ml of LB was monitored at 37 °C in a Bioscreen C machine (Growth Curves USA). MIC values were defined as the minimal antibiotic concentration for more than 90% growth inhibition after 15 h. For MIC and MBC assays adhering to clinical standards[60], an initial inoculum of 5 × 10$^5$ cells was incubated with increasing concentrations of Rif in Mueller–Hinton broth at 37 °C for 18 h. The MIC was determined where at least 90% of bacterial growth is inhibited based on OD600. The MBC was determined as the lowest concentration of antibacterial agent that reduces the viability of the initial bacterial inoculum by at least 99.9% after 18 h at 37 °C.

### Viability assays

Overnight cultures of the indicated strains grown in LB were diluted 1,000-fold and grown to an OD600 of 0.2 in 2 ml of medium before the addition of the indicated concentration of Rif (Sigma), ampicillin (Sigma), chloramphenicol (Sigma), tetracycline (Sigma), 2,2′-dipyridyl (Sigma) or thiourea (Sigma). After 1 h of treatment, cells were washed twice with an equal volume of LB. Serial dilutions of the washed cells were made and either plated on LB agar or placed in a Bioscreen C machine to monitor growth. For thymidine supplementation experiments, LB agar was supplemented with the indicated concentration of thymidine (Sigma) and 5FU (Sigma). For replication block experiments, strains were constructed in the PC2 background carrying a temperature-sensitive allele of the DNA replication protein dnaC[39,42]. Cells were grown as indicated to an OD600 of 0.2 at 30 °C before being shifted to 42 °C for 1.5 h. Cells were then added to media with 80 μg ml$^{-1}$ Rif prewarmed to 42 °C for 30 min before being washed and plated on LB agar. Viability of cells was calculated by normalizing the number of treated to untreated colonies.

### In vitro RNAP runoff assay

The *rpoB* mutant was cloned onto the PVS10 vector for recombinant expression and purification of *E. coli* core RNAP as previously described[61]. Than 3 pmol of RNAP core was mixed with equimolar σ[70] and template DNA with sequence: tccagatcccgaaaatttatcaaaaagagtattgacttaaagtc taacctataggatacttacagccATCGAGAGGGCCCACGGCGAACAGCCAACCC AATCGAACAGGCCTGCTGGTAATCGCAGGCCTTTTTTATTTGGATCCCC GGGTA (capital letters denote transcribed sequence) in 20 μl TB50 buffer (20 mM Tris-HCl pH 8; 10 mM MgCl$_2$, 50 mM NaCl, 0.03% Igepal-60) and incubated 5 min at 37 °C. Assembled transcription complexes were then treated with 2 μl of Rif (20 μg ml$^{-1}$) and incubated for 5 min at 37 °C before being transferred to 10 μl of NeutrAvidin beads

(Fisher) and shaken for 5 min at 25 °C. Samples were washed four times with 1 ml of TB50. Adenosine triphosphate (ATP), guanosine triphosphate (GTP), uridine triphosphate (UTP) and 5 µM cytidine triphosphate (CTP) premixed with CTP-$\alpha$P$^{32}$ was added to 1 mM and incubated at 37 °C. Then 10 µl aliquots were taken after 3, 5, 10, 20, 30 or 40 min and mixed with SB (1× tris-borate-EDTA (TBE); 20 mM EDTA; 8 M urea; 0.025% xylene cyanol, 0.025% bromophenol blue), heated for 5 min at 100 °C and loaded on a 30% (20 × 20 cm$^2$) (19:1) polyacrylamide gel with 7 M urea and TBE prerun for 6 min. The gel was run for 50 min at 50 W before being transferred to an X-ray film for overnight exposure. Gel intensity was quantified using GelQuant.NET.

### ChIP–seq

Overnight cultures of *rpoB* wild-type *rpoC*-10X-His and *rpoB* T525D *rpoC*-10X-His were diluted 1,000-fold and grown to an OD600 of 0.2 in 25 ml of LB at 37 °C before the addition of 80 µg ml$^{-1}$ Rif. After treatment for 1 h, cells were washed once in an equal volume of LB. Cells were allowed to recover for 1 h in LB at 37 °C before crosslinking with 1% formaldehyde at 37 °C for 20 min. A final concentration of 250 mM glycine was added to quench the reaction. Cells were collected by centrifugation at ×5,000*g* and washed twice with ice cold 1 phosphate buffered saline (PBS) before being resuspended in 1 ml of lysis buffer (10 mM Tris pH 8.0, 100 mM NaCl, 1 mM EDTA, 1 mM EGTA, 0.1% sodium deoxycholate, 0.5% *N*-lauroylsarcosine) plus 2 mg ml$^{-1}$ lysozyme with protease inhibitor (Roche) and incubated at 37 °C for 30 min. DNA was sheared using an ultrasonicator Covaris M220 on a 10 s on/10 s off cycle for a total of 50 cycles. Supernatant was incubated with 5 µg ml$^{-1}$ 6X-His-tag antibody (Proteintech) and incubated at 4 °C overnight.

Then 50 µl per sample of Protein A/G beads were pre-equilibrated with 1 ml of ice cold 1× PBS +0.5% bovine serum albumin (BSA) and incubated with samples for 0.5 h at 4 °C. Samples were washed five times in 1 ml of RIPA buffer (50 mM HEPES pH 7.5, 250 mM LiCl, 1 mM EDTA, 1% NP40, 0.7% sodium deoxycholate) and given a final wash in 1× TE (100 mM Tris-Cl pH 8, 10 mM EDTA pH 8). Complexes were uncrosslinked in 250 µl of 1× TE + 4 µl of RNAseA for 1 h at 37 °C and then left overnight at 65 °C with 1% sodium dodecyl sulfate (SDS) and 1 mg ml$^{-1}$ proteinase K. DNA was purified with ChIP Clean and Concentrate (Zymo). ChIP experiments were done in duplicate.

For sequencing, sample libraries were prepared using the NEBNext ChIP–seq library kit (NEB) according to the manufacturer's instructions. Libraries were checked on TapeStation 2200 (Agilent) for quality control. Samples were sequenced on NextSeq 2000 (Illumina). Bowtie[62] and Deeptools[63] were used for alignment and analysis, respectively.

### Analysis of DNA fragmentation

For DNA fragmentation experiments, cells were diluted 1,000-fold from overnight cultures and grown to an OD600 of 0.2 in 2 ml of medium before the addition of 80 µg ml$^{-1}$ Rif or 125 ng ml$^{-1}$ norfloxacin. After 1 h of treatment, cells were washed twice with an equal volume of LB before being fixed with 4% formaldehyde for terminal deoxynucleotide TUNEL. Labelling of DNA fragments was performed using the Apo-Direct Kit (BD Bioscience) following manufacturer's instructions. Samples were collected with a FACSCalibur flow cytometer (BD Biosciences) and at least 50,000 cells were collected for each sample. The percentage of TUNEL positive cells for a given condition is the percentage of cells exceeding the signal detected in more than 99% of untreated cells.

### SLR-qPCR

The number of DSBs in the genomic DNA was determined using SLR-qPCR[38]. SLR-qPCR can be used for detection of any lesion or strand break that creates a barrier to amplification by DNA polymerase[64]. To quantify DSBs in genomic DNA, indicated strains were inoculated in LB. Overnight cultures of indicated strains were diluted 1,000-fold in LB and allowed to grow to an OD600 of 0.2 at 37 °C. Cells were treated either with Rif (80 µg ml$^{-1}$) or the carrier solvent dimethyl sulfoxide

(DMSO) for 1 h at 37 °C. Cells were harvested by centrifugation and washed twice with PBS. Genomic DNA was isolated using a Monarch Genomic DNA isolation kit (NEB).

The reaction mixture contained 1× SYBR Green mix (Applied Biosystems), 500 nM of each primer and 1 ng of template DNA in a total volume of 20 µl per well. Two sets of primers (Supplementary Table 3), yielding a short and a long amplicon provide data representing the total amount of template (an internal normalization control) and undamaged DNA, respectively. The two primer pairs had comparable efficiency of amplification and yielded a single PCR product as judged by agarose gel analysis. Data analysis was done as described[38] using a modified version of the 2−ΔΔCT method. DSB density was measured using genomic DNA from untreated cells as reference. DNA damage was calculated as DSBs per 10 kilobase DNA.

For replication block experiments, a dual plasmid system expressing dCas9 and a guide RNA targeting the oriC was introduced to cells of interest as previously reported[41]. Cells were grown to an OD600 of 0.2 at 37 °C as described above, and dCas9 was induced with 200 ng ml$^{-1}$ Tc for 30 min before adding Rif.

### Measurement of reactive oxygen species

Cells were diluted 1,000-fold from overnight cultures and grown to an OD600 of 0.2 in 2 ml of medium before the addition of 80 µg ml$^{-1}$ Rif or 10 µg ml$^{-1}$ ampicillin. Cells were incubated with antibiotic for 30 min before the addition of 10 mM carboxy-H2DCFDA (Thermo), after which cells were incubated for another 30 min. Cells were then washed with an equal volume of PBS, normalized for cell number and transferred to an opaque 96-well plate, after which fluorescent intensity was measured with a microplate reader. A blank well loaded with an equal volume of PBS was used to control for background fluorescence.

### Western blot

Cells were diluted 1,000-fold from overnight cultures and grown to an OD600 of 0.2 in 2 ml of medium before the addition of 80 µg ml$^{-1}$ Rif for 1 h. Cells were then collected, resuspended in lysis buffer (10 mM Tris-HCl pH 7.5, 4% SDS), mixed with 4× lithium dodecyl sulfate (LDS) sample buffer (Invitrogen), and boiled at 95 °C for 10 min before loading on a NuPAGE 4–12% Bis-Tris precast gel (Thermo) for electrophoresis at 200 V for 40 min. Gel was then semidry transferred to a polyvinylidene difluoride (PVDF) membrane (Millipore) and incubated with the specified antibodies at 4 °C overnight in SuperBlock T20 blocking buffer (Thermo) before development the following day. Antibodies used were *E. coli* RNAP beta Monoclonal Antibody clone 8RB13 mouse mAB at 1:2,000 (663905; Biolegend) and LexA Antibody (E-7) at 1:200 (sc-365999; Santa Cruz). Gel intensity was quantified using GelQuant.NET.

### RT–qPCR experiments

RNA was prepared from cells treated with 80 µg ml$^{-1}$ Rif for 1 h using Trizol and Direct-zol RNA microprep (Zymo). RT–qPCR was done with 100 ng of RNA with the Luna RT–qPCR kit (NEB) and Quantstudio 7 (Applied Biosystems) using primers targeting rpsL as a control (Supplementary Table 5).

### RNA-sequencing library preparation and analysis

RNA-sequencing libraries were prepared from mid log cells grown in LB with the NEB Ultra II Directional RNA Library Prep Kit for Illumina (NEB, E7760S). Reads were trimmed with Cutadapt and then aligned to the *E. coli* genome with Bowtie2 (ref. 62). DESeq2 (ref. 65) was used to analyse differential expression.

### Metabolomics

Mid log cells grown in LB were spun down, washed with PBS and resuspended in 1 m; of dry-ice-cooled extraction buffer (50% methanol, 1% formic acid, 10 µM labelled UTP;13C9,15N2) optimized for UTP extraction. Cells were transferred to Beadblaster (Benchmark Scientific) tubes

and homogenized with ten cycles (30 s on, 30 s off). Homogenized samples were transferred to a 2 ml glass vial and vortexed with 0.4 ml of HPLC grade water and 0.8 ml of chloroform. Samples were then incubated at 4 °C for 30 min and the supernatant was separated and dried in a SpeedVac (Thermo Scientific) before being resuspended in 40 μl of HPLC grade water. The resuspended sample was then centrifuged to remove debris and transferred into a 250 μl glass insert for LC–MS analysis at the NYU metabolomics core.

## Transcription kinetics

Transcription elongation rates were measured with a multiprobe (Supplementary Table 3) qPCR assay along the *lac* operon as previously described[43]. *E. coli* cells were grown to OD600 of 0.4 before induction by 1 mM isopropyl beta-D-thiogalactopyranoside (IPTG). Following induction, aliquots were withdrawn at 10 s intervals into a tube containing stop solution (60% EtOH, 2% phenol, 10 mM EDTA) precooled to −20 °C. RNA was extracted using Trizol (Invitrogen) and the Direct-zol RNA Microprep kit (Zymo) according to manufacturer's instructions. Then 100 ng of total RNA was converted to complementary DNA using Superscript IV reverse transcriptase (ThermoFisher) and RNaseOut (ThermoFisher). Real-time qPCR amplification was performed with SYBR Green (Invitrogen) and Quantstudio 7 (Applied Biosystems). Data analysis was done as previously described[43].

## NETseq

NETseq libraries were prepared and analysed as previously described[4], with modifications to the pause calling algorithm. Pauses were called using 100-nucleotide sliding windows with the condition that the pause is the maximum signal within the window and the signal is 2 s.d. above the window mean.

## Evolutionary sequence analysis

The KEGG orthology database[49] was mined for orthologues of *E. coli rpoB* and analysed in a similar manner to work previously done with *rpoD*[28]. In brief, orthologues were filtered for bacterial proteins that were of similar length to *E. coli rpoB* yielding 1,342 sequences. Next, sequences were aligned with Clustal Omega[66] and positional conservation was calculated by Jensen–Shannon divergence[50]. Orthologue variants within the *rpoB* binding site were determined by comparison to *E. coli* MG1655 *rpoB* and their number and frequency were calculated.

## Reporting summary

Further information on research design is available in the Nature Portfolio Reporting Summary linked to this article.

## Data availability

Sequencing data is available through the NCBI sequence read archive (SRA) under PRJNA992395. Source data are provided with this paper.

## Code availability

Code for analysing MAGE-seq datasets was adapted from previous work[28].

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

**Acknowledgements** We thank P. Zappile from the NYU Genome Technology Center, which is partially supported by the Cancer Center Support grant no. P30CA016087, at the Laura and Isaac Perlmutter Cancer Center. We thank Y. Siu and D. Jones from the NYU Metabolomics Core Resource Laboratory. We thank K. Alzoubi for help with data visualization. This work was supported by the Blavatnik Family Medical Foundation (E.N.); Howard Hughes Medical Institute (E.N.); NIH grant R01GM126891 (E.N.); NYU Medical Scientist Training Program (K.B.Y.); and Public Health Service Institutional Research Training Award no. T32 AI007180 (K.B.Y.).

**Author contributions** K.B.Y. and A.R. performed the MAGE experiments presented in Figs. 1–3 and Extended Data Figs. 1–3. K.B.Y. performed the data analysis and visualization of MAGE data and evolutionary analysis presented in Figs. 1–5 and Extended Data Figs. 1–3. K.B.Y., A.R. and M.C. prepared the strains and performed the validation experiments presented in Figs. 3 and 4 and Extended Data Figs. 5–9. C.M. performed the flow cytometry experiments presented in Fig. 3. M.G. performed SLR-qPCR and transcription kinetic experiments presented in Figs. 3 and 4 and Extended Data Fig. 7. Z.H. purified proteins and V.E. performed in vitro transcription assays presented in Extended Data Fig. 4. Y.S. performed the ChIP–seq experiments presented in Fig. 3. T.N. performed western blots presented in Extended Data Fig. 6. E. Nirenstein performed the MIC assays presented in Extended Data Fig. 9. I.S. performed next-generation sequencing. K.B.Y, A.R. and E. Nudler designed the project and wrote the manuscript with input from all authors. A.R. and E. Nudler supervised the project. All authors discussed the results.

**Competing interests** The authors declare no competing interests.

**Additional information**
**Correspondence and requests for materials** should be addressed to Aviram Rasouly or Evgeny Nudler.

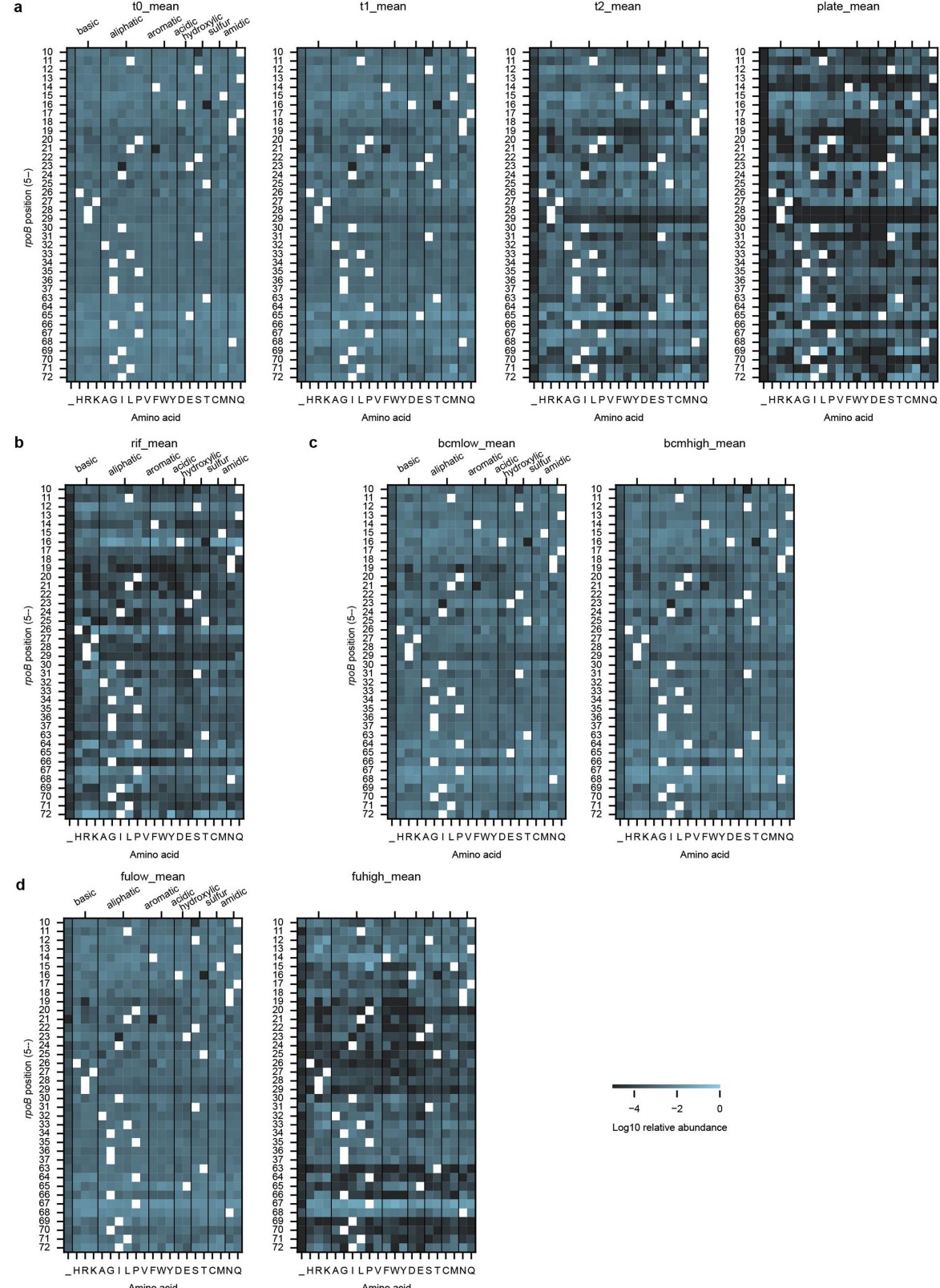

**Extended Data Fig. 1 | Mutant abundance following MAGE-seq screening of rifampicin binding site mutants. a-d**, Representative heatmaps of relative mutant abundances from MAGE-seq screening under the following conditions: **a**, timepoints 0–2 ($T_{0-2}$) and plate. $T_0$ represents the initial MAGE-seq mutant pool. $T_1$ and $T_2$ are after subsequent 100 fold dilutions, allowing cells to reach the same OD before harvest. Plate represents MAGE-seq libraries prepared from mutants passaged on solid medium (LB agar). **b**, mutant abundance

following screening with rifampicin. **c**, mutant abundance following screening with 15 (bcmlow) or 30 ug/mL (bcmhigh) of bicyclomycin. **d**, mutant abundance following screening with 1 (fulow) or 5 ug/mL (fuhigh) of 5-fluorouracil. Rows correspond with *rpoB* positions 510–537 and 563–572; columns correspond to all 20 amino acids along with stop codons. White squares denote wild-type residues. Each heatmap displays values from the mean of $n$ = 3 biologically independent MAGE-seq experiments.

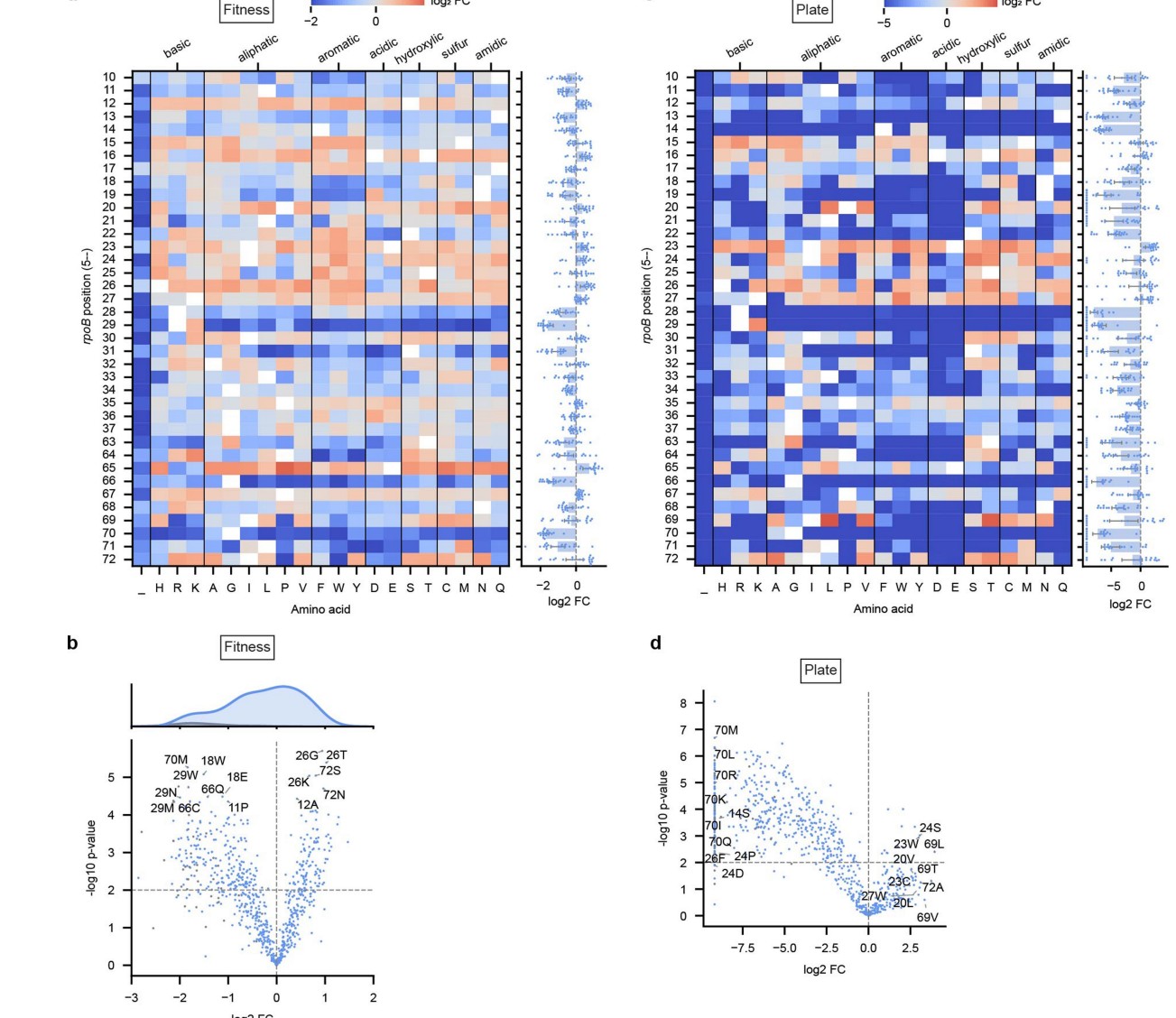

**Extended Data Fig. 2 | Fitness of rifampicin binding site mutants. a**, Fitness landscape of *rpoB* mutants comparing timepoint 1 to timepoint 0. **b**, Volcano plot of mutant pool fitness. Stop codons are in gray. Kernel density estimate plot of mutant fitness distribution is displayed above. **c**,**d**, Fitness landscape and volcano plot of *rpoB* mutants comparing those that form colonies on agar plates to timepoint 0. For (a) and (c) rows correspond with *rpoB* position and columns correspond with all 20 possible amino acid substitutions, grouped by property. White squares denote wild-type residues. Barplot of average positional fold change is presented right of the heatmap. All MAGE-seq experiments were conducted in *n* = 3 biologically independent replicates. *P* values are calculated using the independent t-test (two-sided). Error bars denote 95% confidence intervals.

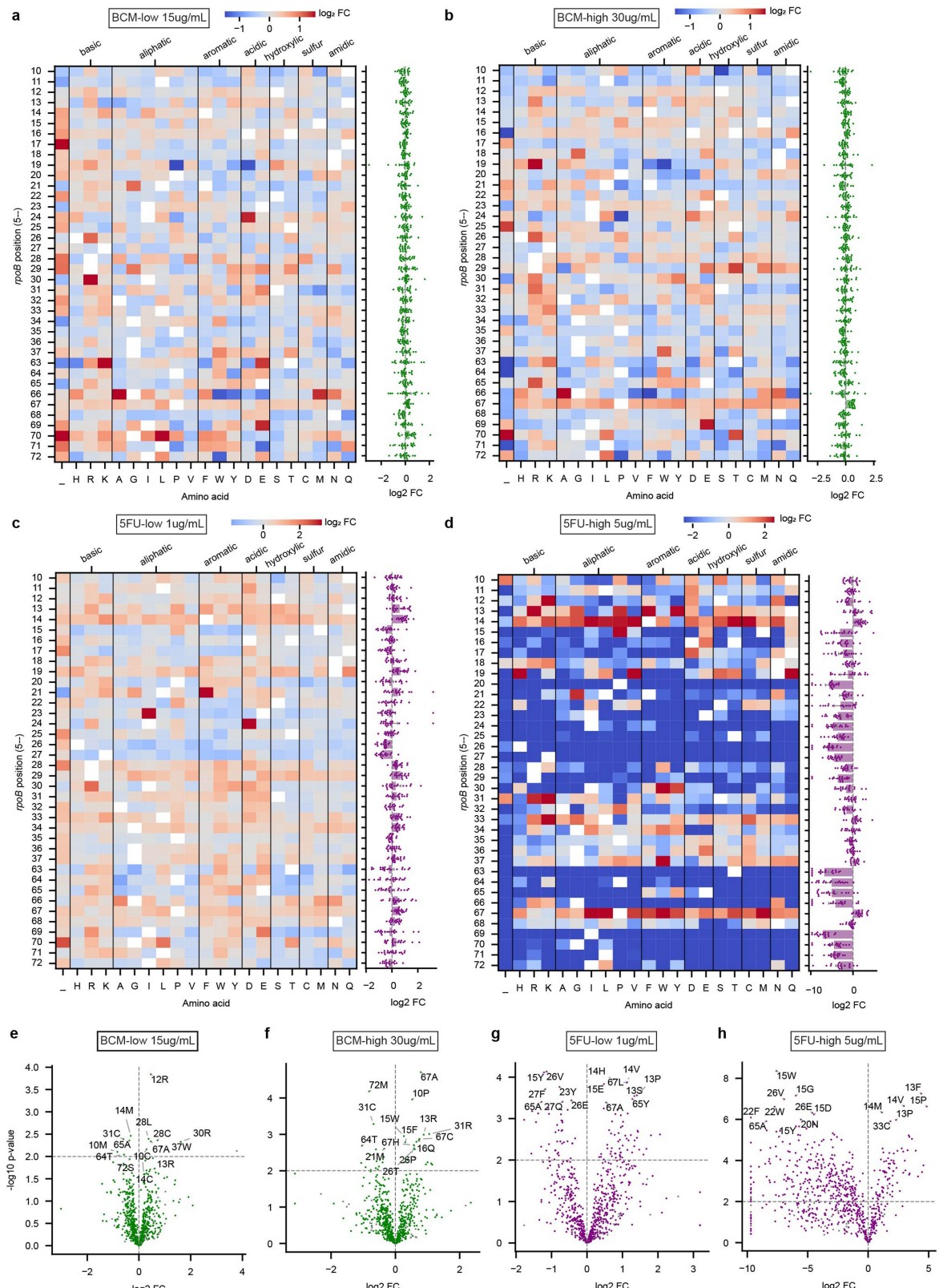

**Extended Data Fig. 3 | Resistance and sensitivity landscape of rifampicin binding site mutants towards bicyclomycin and 5-fluorouracil. a–d**, Heatmap rows correspond with *rpoB* position and columns correspond with all 20 possible amino acid substitutions, grouped by property. White squares denote wild-type residues. Barplot of average positional fold change is presented right of the heatmap. **e–h**, Volcano plots of data in (a–d). All MAGE-seq experiments were conducted in *n* = 3 biologically independent replicates. *P* values are calculated using the independent t-test (two-sided). Error bars denote 95% confidence intervals.

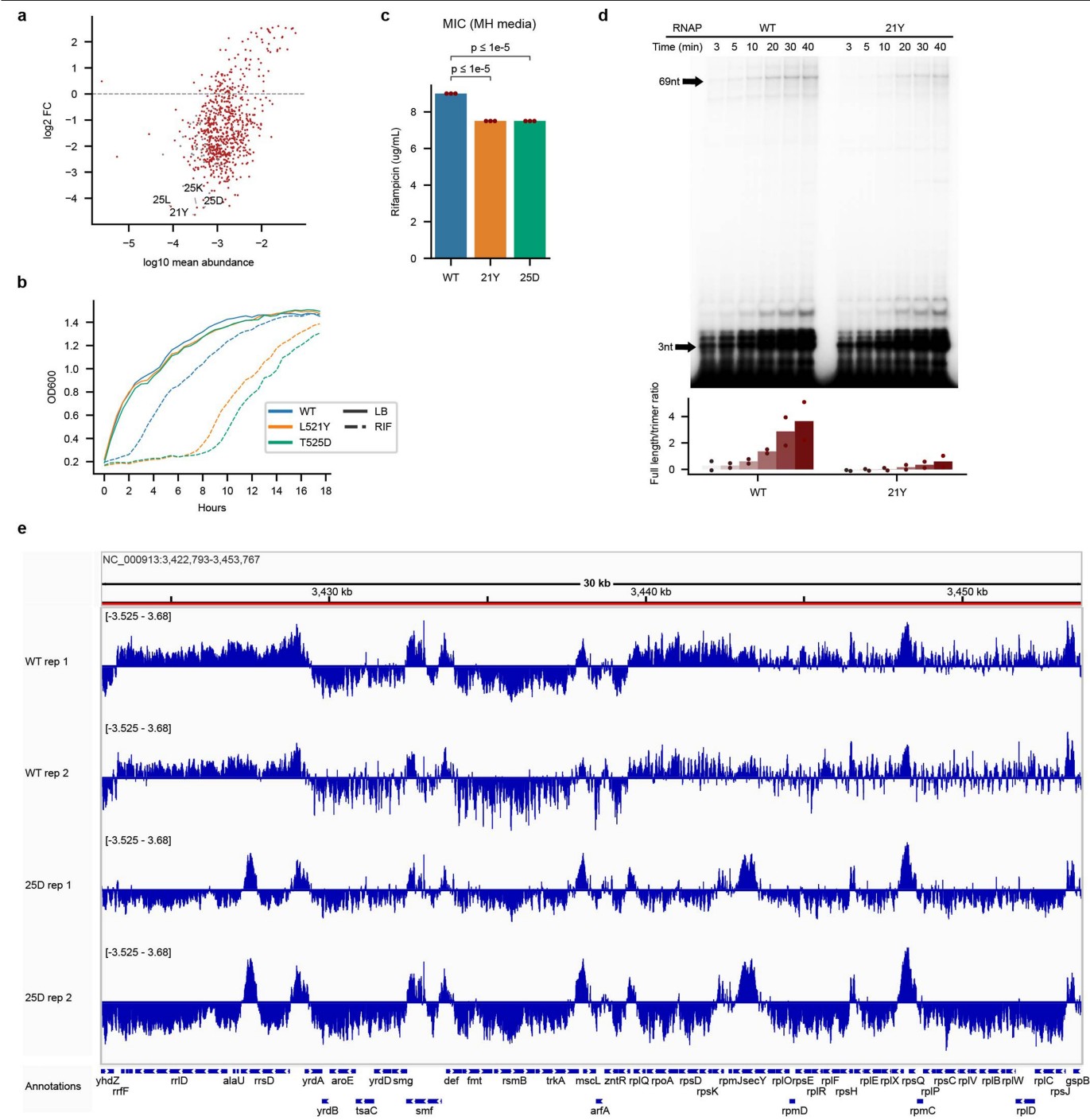

**Extended Data Fig. 4 | Characteristics of rifampicin hypersensitive mutants of RNA polymerase. a**, MA plot of MAGE mutant pool sensitivity and resistance to rifampicin. Stop codons are in gray. **b**, Growth curves of hypersensitive mutant recovery after 1 hr of treatment with 80 ug/mL rifampicin (*n* = 2). **c**, MIC of hypersensitive strains (*p* < 1e-5, 1e-5 from left to right). An initial inoculum of 5 × 10⁵ cells was incubated with increasing concentrations of rifampicin in Mueller-Hinton (MH) broth at 37C for 18 h. **d**, *In vitro* transcription recovery reaction of wild-type and *rpoB* L521Y RNAP following rifampicin treatment and wash. Location of the trimer and terminator (69nt) are denoted

with arrows. Barplot of the intensity ratio of the terminator/trimer at each timepoint is provided below as the average of two independent experiments. **e**, Representative genome viewer of chromatin immunoprecipitation sequencing (ChIP-seq) signal from wild-type and *rpoB* T525D RNAP normalized to input. Cells were collected after 1 h recovery in LB from 1 h of treatment with 80 ug/mL rifampicin. Bar plots and growth curves represent the mean of *n* = 3 biologically independent experiments unless otherwise noted, with error bars denoting 95% confidence intervals. *P* values are calculated using the independent t-test (two-sided).

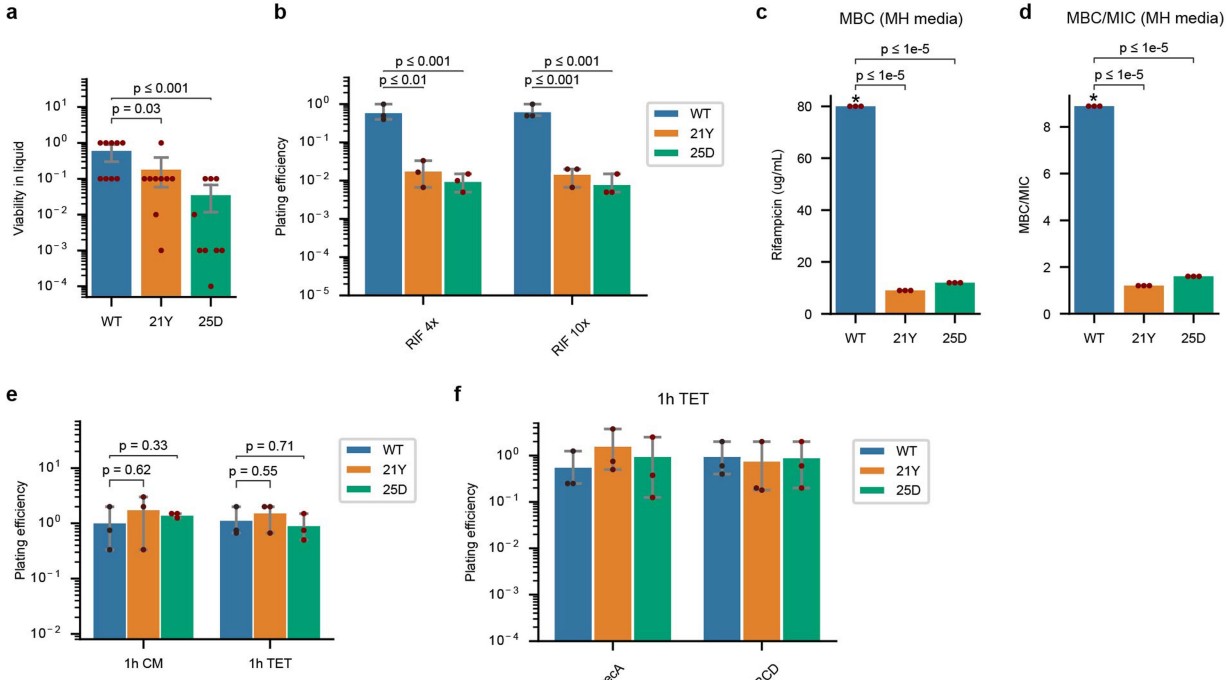

**Extended Data Fig. 5 | Bactericidal effect of rifampicin in hypersensitive mutants. a**, Viability in liquid of hypersensitive mutants following 1 h of treatment with rifampicin. Cells in exponential phase were treated with 80 ug/mL of rifampicin for 1 h at 37C before being washed, serially diluted, and measured for cell growth ($n = 9$, $p = 3.1e$-$2$, $3.1e$-$4$ from left to right). **b**, Plating efficiency of hypersensitive mutants following 1 h of treatment with MIC normalized concentrations of rifampicin ($p = 2.6e$-$3$, $6.0e$-$4$, $9.1e$-$4$, $5.0e$-$4$). Cells in exponential phase were treated with 4x or 10x their respective minimal inhibitory concentration (MIC) of rifampicin before being washed, serially diluted, and plated. **c**, Minimal bactericidal concentration (MBC) of hypersensitive strains ($p < 1e$-$5$, $1e$-$5$). The MBC was determined as the lowest concentration of antibacterial agent that reduces the viability of the initial bacterial inoculum by ≥99.9% after 18 h at 37C. Rifampicin was tested up to the

concentration of 80 ug/mL indicated by the asterisk. **d**, MBC/MIC ratio determined for rifampicin hypersensitive strains ($p < 1e$-$5$, $1e$-$5$). **e**, Plating efficiency of hypersensitive mutants following 1 h of treatment with rifampicin, chloramphenicol (CM), and tetracycline (TET) ($p = 6.2e$-$1$, $3.3e$-$1$, $5.5e$-$1$, $7.1e$-$1$). Cells in exponential phase were treated with 80ug/mL of rifampicin, 20 ug/mL of chloramphenicol, or 20ug/mL of tetracycline for 1 h at 37C before being washed, serially diluted, and plated on LB. **f**, Plating efficiency of hypersensitive mutants in *recA* or *recBCD* knockout strains after 1hr treatment with 20 ug/mL tetracycline at 37C. Cells in exponential phase were treated with 20 ug/mL of tetracycline for 1 h at 37C before being washed, serially diluted, and plated on LB. Bar plots represent the mean of $n = 3$ biologically independent experiments unless otherwise noted, with error bars denoting 95% confidence intervals. *P* values are calculated using the independent t-test (two-sided).

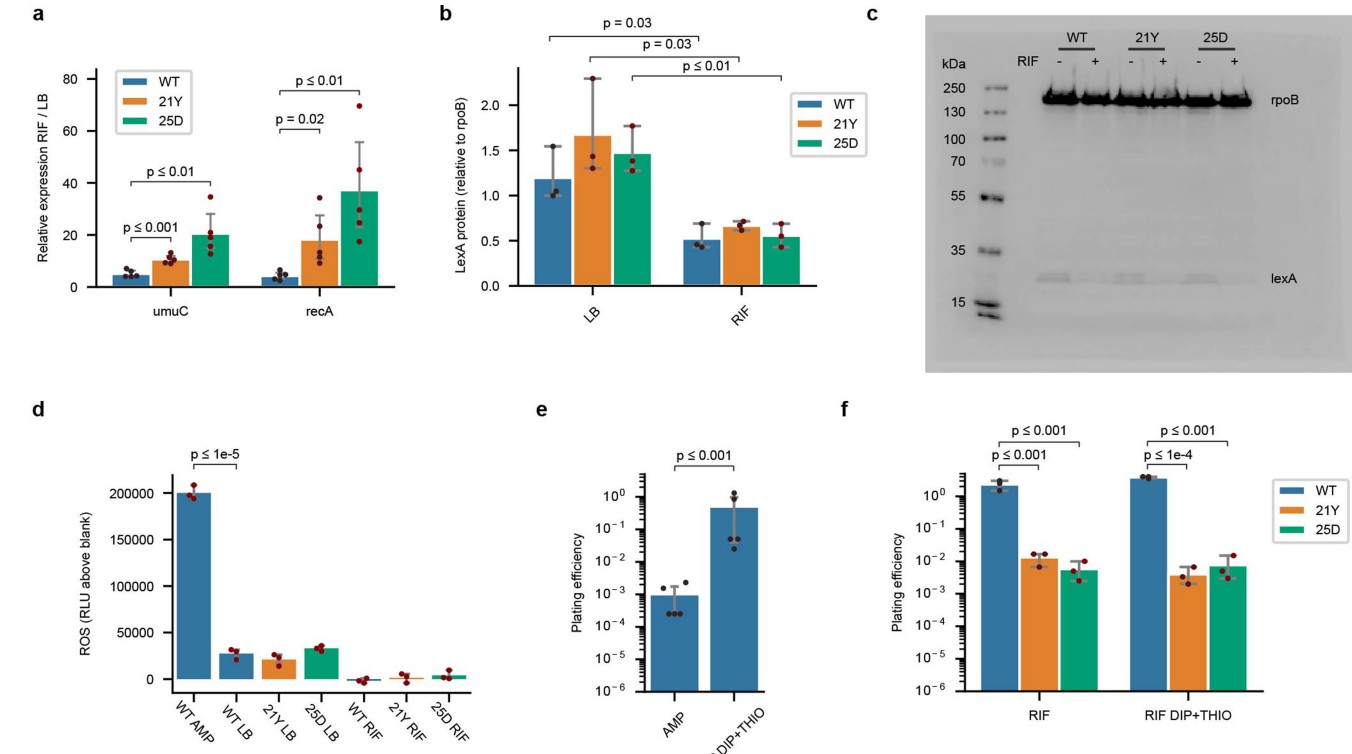

**Extended Data Fig. 6 | SOS response and ROS in rifampicin treated cells.**
**a**, Relative expression of SOS response genes in response to 1hr treatment of
80 ug/mL rifampicin at 37C by RT-qPCR. Data was normalized to expression
of *rpsL* and then between Rif and untreated conditions ($n = 5$, $p = 4.7e-4$, $3.7e-3$,
$1.8e-2$, $7.5e-3$ from left to right). **b**, *LexA* relative protein levels measured by
western blot in exponentially growing hypersensitive mutants following 1 hr
treatment of 80ug/mL rifampicin at 37C ($p = 2.5e-2$, $3.2e-2$, $5.4e-3$). Data is
normalized to *rpoB* relative protein levels. **c**, Representative western blot of
data in (b). **d**, Reactive oxygen species (ROS) of cells treated with ampicillin
or rifampicin, measured by carboxy-H2DCFDA ($p = 5.8e-6$). Exponentially

growing cells were treated for 1hr with 80 ug/mL rifampicin or 10 ug/mL
ampicillin at 37C. Carboxy-H2DCFDA was added 30 min prior to sample
collection. **e**, Plating efficiency of wild-type cells treated with 10 ug/mL
ampicillin with or without the addition of 500 uM 2,2' dipyridyl and 150 mM
thiourea ($n = 5$; $p = 3.9e-4$). **f**, Plating efficiency of hypersensitive mutants
following 1 h of treatment with 80 ug/mL rifampicin with or without the
addition of 500 uM 2,2' dipyridyl and 150 mM thiourea ($p = 1.5e-4$, $1.7e-4$,
$3.8e-5$, $1.7e-4$). Bar plots represent the mean of $n = 3$ biologically independent
experiments unless otherwise noted, with error bars denoting 95% confidence
intervals. $P$ values are calculated using the independent t-test (two-sided).

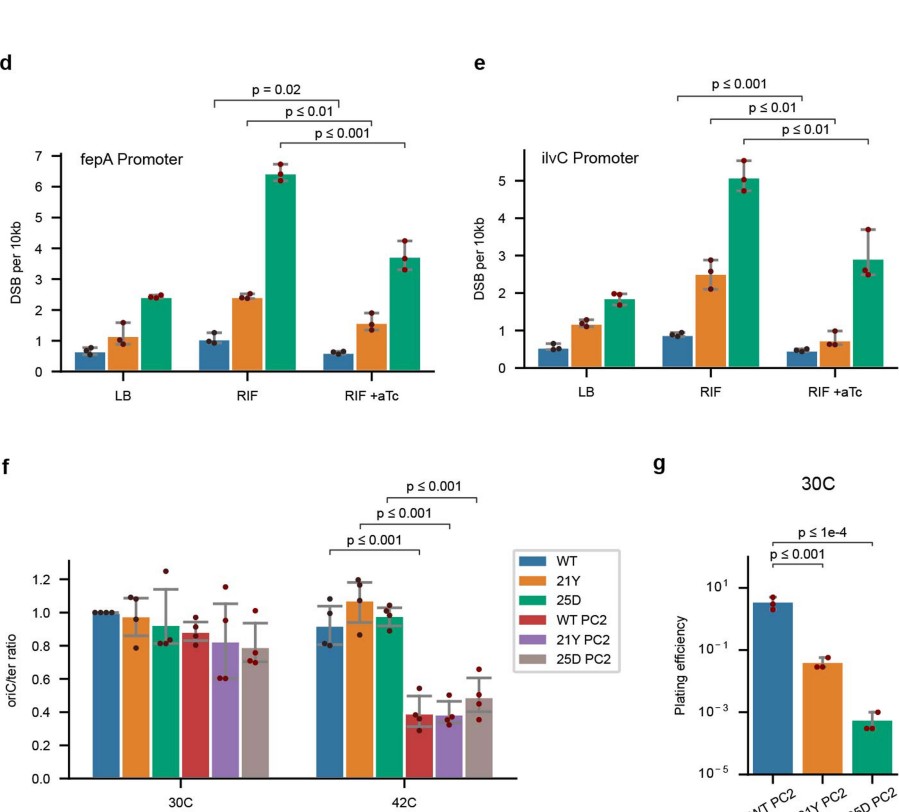

**Extended Data Fig. 7 | Role of replication in the bactericidal effects of rifampicin. a**, Schematic of semi long range quantitative PCR (SLR-qPCR) used to detect double strand breaks (DSB). Two sets of primers, yielding a short and a long amplicon provide data representing the total amount of template (an internal normalization control) and undamaged DNA respectively. DSB density is measured using genomic DNA from untreated cells as reference. DNA damage is calculated as DSBs per 10 kb DNA. **b**, Schematic of SLR-qPCR primer design around the promoter and gene-body of selected genes. **c**, Schematic for replication block experiments, a dual plasmid system expressing dCas9 and a guide RNA targeting the oriC was introduced to cells of interest. dCas9 prevents binding of replication initiation proteins. dCas9 was induced with 200 ng/mL of aTc for 30 min prior to adding Rif. **d**,**e**, Number of DSB per 10

kilobases at the *fepA* and *ilvC* promoter measured using SLR-qPCR following 1 hr treatment with 80ug/mL rifampicin with or without replication (*p* = 1.5e-2, 7.6e-3, 9.9e-4, 3.0e-4, 2.3e-3, 8.6e-3 from left to right). Replication was blocked using dCas9 with a guide RNA targeting the oriC for 30 min prior to adding rifampicin. **f**, qPCR measurement of the oriC/ter ratio of hypersensitive mutants in the PC2 background before and after shifting cells to 42C, normalized to actively replicating wild-type cells at 30C. Cells with a lower oriC/ter ratio are replicating less (*n* = 4; *p* = 9.3e-4, 1.9e-4, 4.8e-4). **g**, Plating efficiency of hypersensitive mutants in PC2 at the replication permissive temperature 30C (*p* = 2.2e-4, 5.1e-5). Bar plots represent the mean of *n* = 3 biologically independent experiments unless otherwise noted, with error bars denoting 95% confidence intervals. *P* values are calculated using the independent t-test (two-sided).

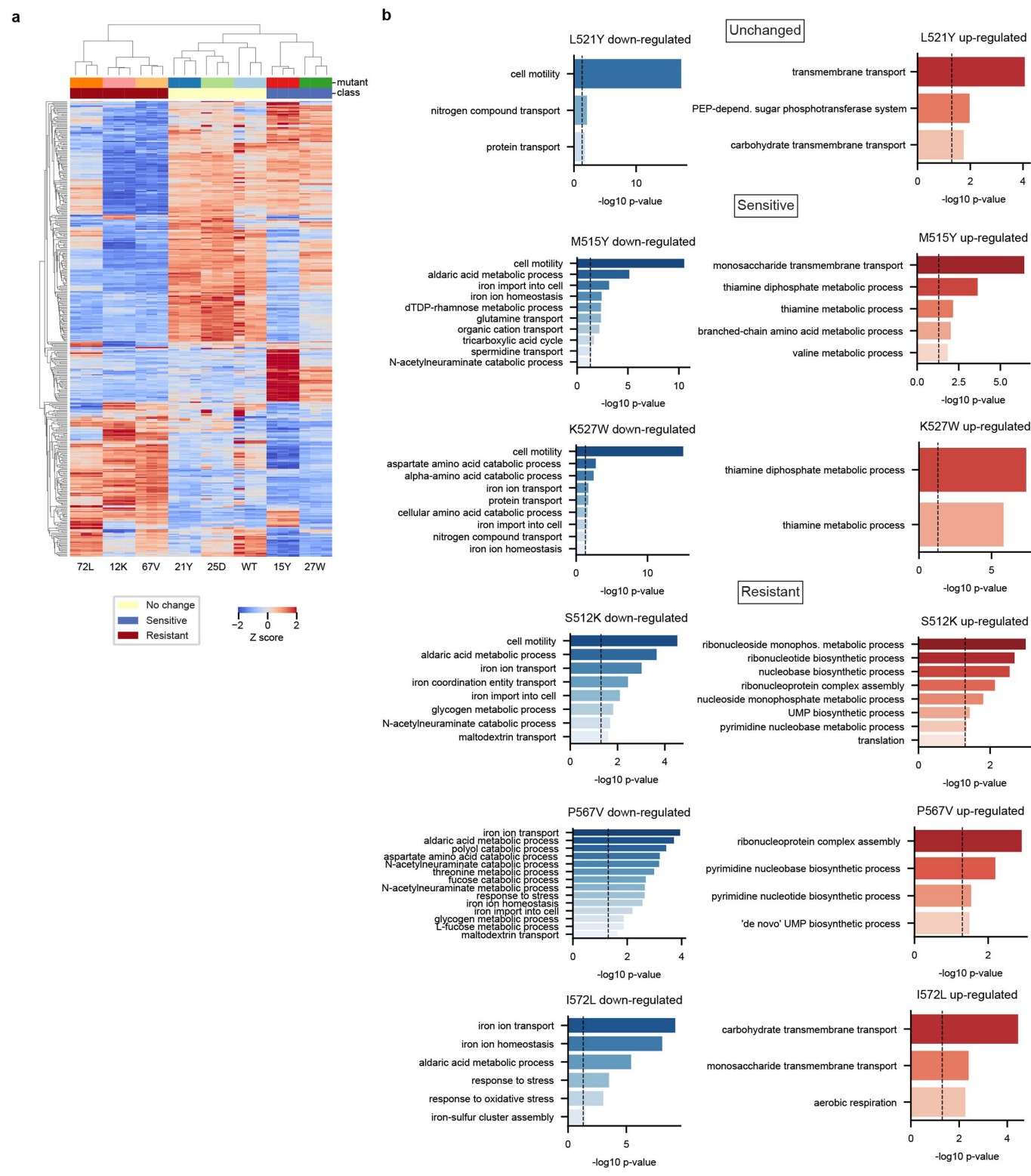

**Extended Data Fig. 8 | Transcriptional profiles of rifampicin binding site mutants. a**, Clustering of RNA expression counts of *rpoB* mutants with increased, decreased, or unchanged sensitivity towards bicyclomycin and 5-fluourouracil. Cells were grown in LB and samples were collected in *n* = 3 biologically independent replicates. Volcano plots from RNAseq analysis is provided in Supplementary Fig. 2. **b**, GO enrichment analysis of differentially regulated genes in rifampicin binding site mutants as compared to wild-type *E. coli*. *P* values are calculated using the one-sided Fishers exact test with Benjamini-Hochberg correction.

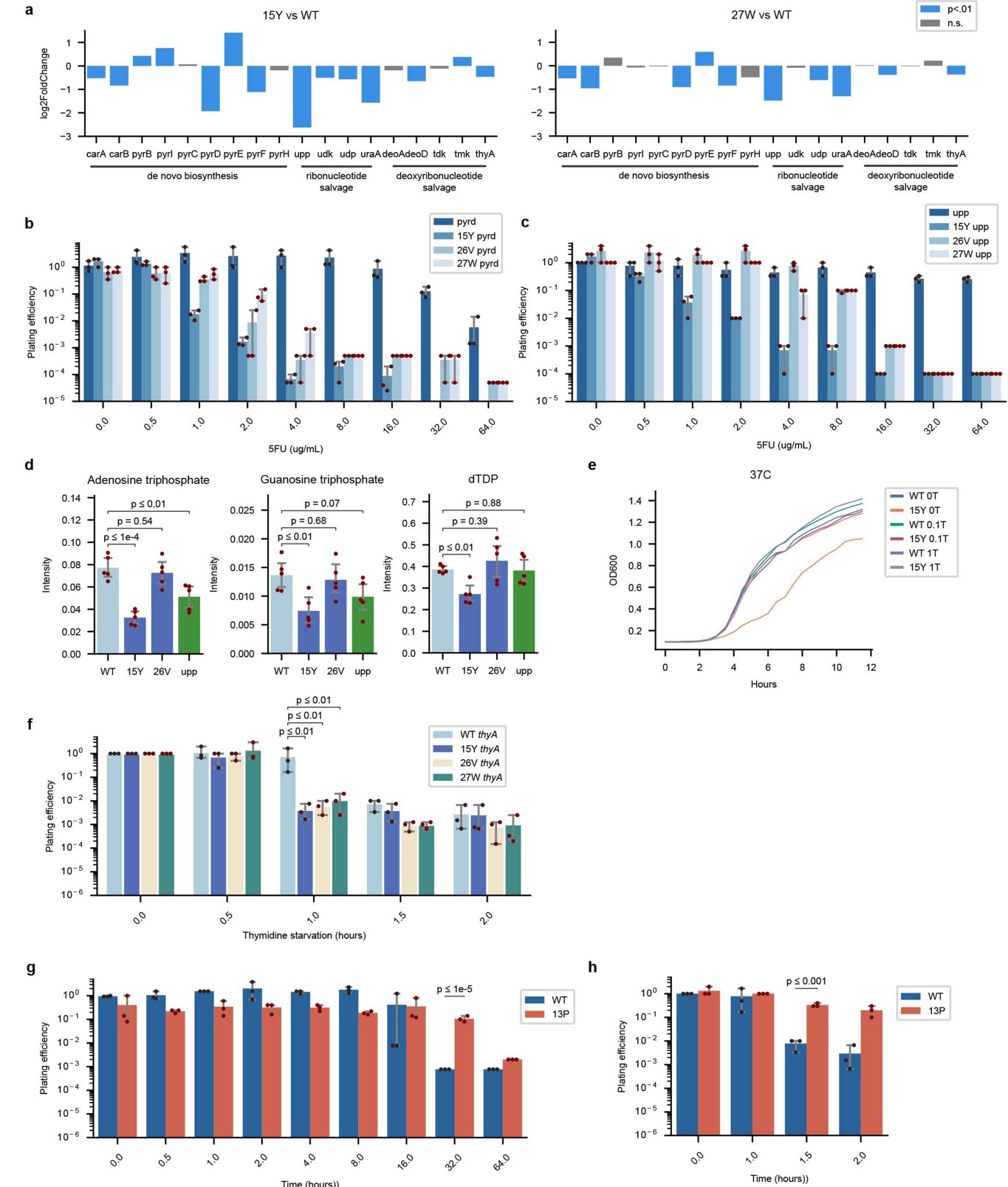

**Extended Data Fig. 9** | See next page for caption.

**Extended Data Fig. 9 | Characteristics of RNAP mutants sensitive or resistant to 5-fluorouracil. a**, Differential gene expression of pyrimidine de novo biosynthesis, ribonucleotide salvage, or deoxyribonucleotide salvage pathways in fast mutants *rpoB* 515Y and 527W compared to wild-type, as measured by RNAseq. Statistically significant differences ($p < 01$) are shown in blue. RNAseq was performed in $n = 3$ biologically independent replicates and two-sided $p$ values were calculated using the Wald test with Benjamini-Hochberg correction. **b,c**, Plating efficiency of 5-fluorouracil sensitive mutants in a background of *pyrD* or *upp*. Cells from overnight cultures were serially diluted and plated on LB agar plates with the indicated concentrations of 5FU in ug/mL. **d**, Purines and dTDP in 5-fluorouracil sensitive rpoB mutants as detected by LCMS ($n = 5$; $p = $ 4.5e-5, 5.4e-1, 6.4e-3, 7.1e-3, 6.8e-1, 7.5e-2, 1.1e-3, 3.9e-1, 8.8e-1 from left to right). **e**, Growth curves of *rpoB* 515Y compared to wild-type cells with the addition of 0, .1, or 1ug/mL of thymidine. Values represent the mean of 2 biological replicates. **f**, Plating efficiency of 5-fluorouracil sensitive mutants in a *thyA* background following thymidine starvation ($p = $ 3.2e-3, 4.2e-3, 9.8e-3). **g**, Plating efficiency of *rpoB* 513P with the indicated concentrations of 5FU in ug/mL ($p = $ 7.2e-6). **h**, Plating efficiency of *rpoB* 513P in a thyA background following thymidine starvation ($p = $ 5.2e-4). Barplots and growth curves represent the mean of $n = 3$ biologically independent experiments unless otherwise noted, with error bars denoting 95% confidence intervals. *P* values are calculated using the independent t-test (two-sided).

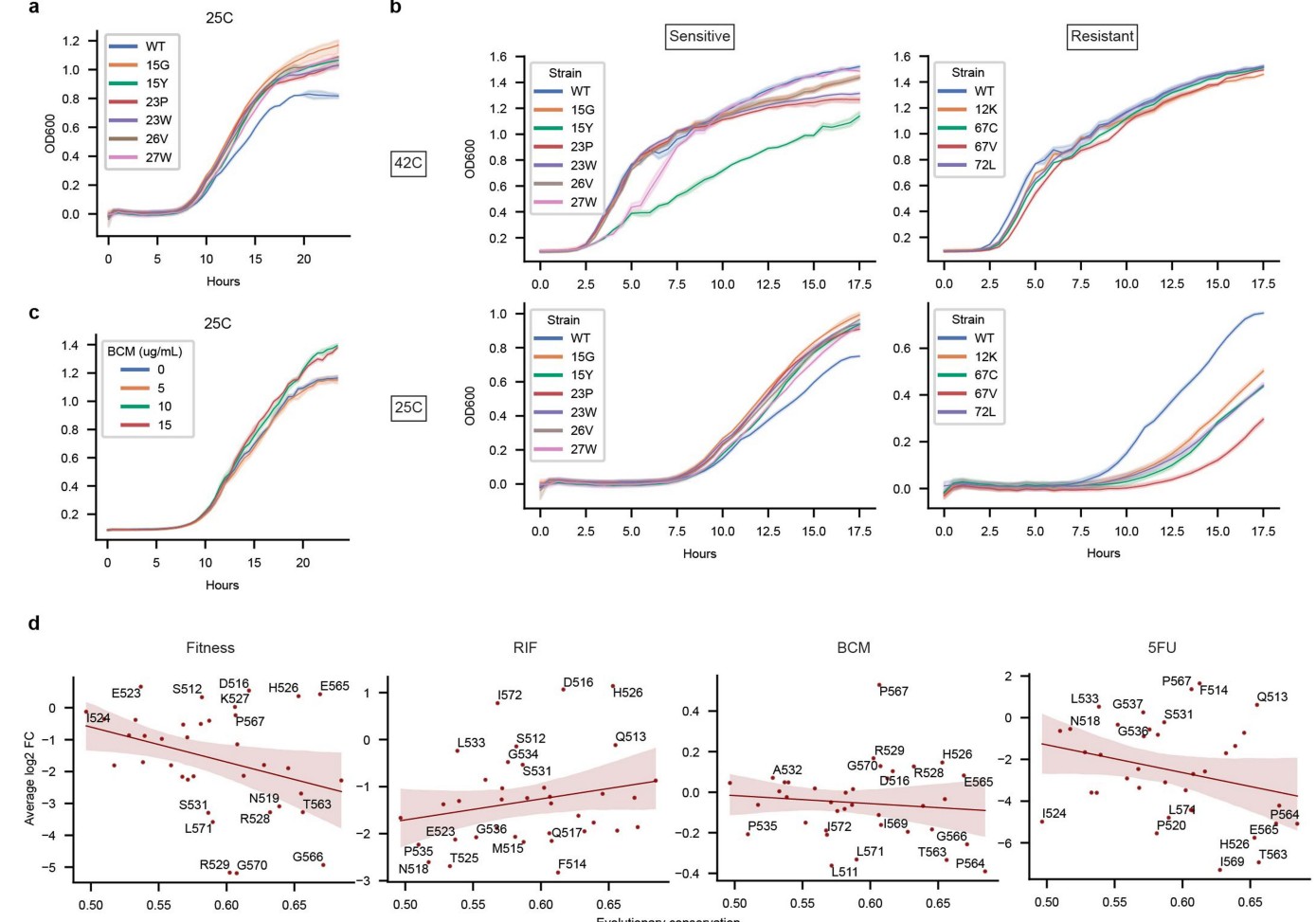

**Extended Data Fig. 10 | Growth characteristics of rifampicin binding site mutants and evolutionary conservation compared to average positional characteristics. a**, Growth curves of 5-fluorouracil sensitive mutants grown at 25 °C. **b**, Growth curves of *rpoB* mutants stratified by 5-fluuorouracil sensitivity at 25C and 42C. **c**, Growth curves of wild-type cells with subinhibitory bicyclomycin, grown at 25 °C. Growth curves represent the mean of *n* = 3 biologically independent experiments unless noted otherwise, with error bars denoting 95% confidence intervals. **d**, Scatterplots of evolutionary conservation and average positional fitness, resistance, or sensitivity for each residue position. Linear regression lines and 95% confidence intervals are overlayed.

# Reporting Summary

## Statistics

For all statistical analyses, confirm that the following items are present in the figure legend, table legend, main text, or Methods section.

| n/a | Confirmed | |
|---|---|---|
| ☐ | ☒ | The exact sample size (*n*) for each experimental group/condition, given as a discrete number and unit of measurement |
| ☐ | ☒ | A statement on whether measurements were taken from distinct samples or whether the same sample was measured repeatedly |
| ☐ | ☒ | The statistical test(s) used AND whether they are one- or two-sided *Only common tests should be described solely by name; describe more complex techniques in the Methods section.* |
| ☐ | ☒ | A description of all covariates tested |
| ☐ | ☒ | A description of any assumptions or corrections, such as tests of normality and adjustment for multiple comparisons |
| ☐ | ☒ | A full description of the statistical parameters including central tendency (e.g. means) or other basic estimates (e.g. regression coefficient) AND variation (e.g. standard deviation) or associated estimates of uncertainty (e.g. confidence intervals) |
| ☐ | ☒ | For null hypothesis testing, the test statistic (e.g. *F*, *t*, *r*) with confidence intervals, effect sizes, degrees of freedom and *P* value noted *Give P values as exact values whenever suitable.* |
| ☒ | ☐ | For Bayesian analysis, information on the choice of priors and Markov chain Monte Carlo settings |
| ☒ | ☐ | For hierarchical and complex designs, identification of the appropriate level for tests and full reporting of outcomes |
| ☐ | ☒ | Estimates of effect sizes (e.g. Cohen's *d*, Pearson's *r*), indicating how they were calculated |

*Our web collection on statistics for biologists contains articles on many of the points above.*

## Software and code

Policy information about availability of computer code

| | |
|---|---|
| Data collection | Sequencing data was collected on the Novaseq 6000 (Illumina) at the NYU Genome Technology Center and an in-house NextSeq 6000 (Illumina). qPCR data was collected on a QuantStudio 7 (Applied Biosystems). Flow cytometry data was collected on a FACSCalibur (BD Biosciences). |
| Data analysis | Sequencing data was analyzed using CutAdapt (vl.18), Bowtie2 (v2.4.1), and deeptools2 (v3.5.l). Software adapted from https://github.com/jiminpark66/MAGEseq (Park et al., 2021) was used for MAGE-seq analysis. For plots, error bars denote 95% confidence intervals and p-values are determined using the independent t-test (two-tailed) unless otherwise noted. FlowJo (vl0.6.0) was used for flow cytometry analysis. Gel intensity was quantified using GelQuant.NET software provided by biochemlabsolutions.com. PyMOL (v2.3.3) was used for visualization. |

For manuscripts utilizing custom algorithms or software that are central to the research but not yet described in published literature, software must be made available to editors and reviewers. We strongly encourage code deposition in a community repository (e.g. GitHub). See the Nature Portfolio guidelines for submitting code & software for further information.

## Data

Policy information about availability of data

All manuscripts must include a data availability statement. This statement should provide the following information, where applicable:

- Accession codes, unique identifiers, or web links for publicly available datasets
- A description of any restrictions on data availability
- For clinical datasets or third party data, please ensure that the statement adheres to our policy

> Sequencing data is available through the NCBI sequence read archive (SRA) under PRJNA992395. Protein Data Base accession number 5UAC was used for visualization.

## Research involving human participants, their data, or biological material

Policy information about studies with human participants or human data. See also policy information about sex, gender (identity/presentation), and sexual orientation and race, ethnicity and racism.

| | |
|---|---|
| Reporting on sex and gender | No human research participants were used in this study. |
| Reporting on race, ethnicity, or other socially relevant groupings | No human research participants were used in this study. |
| Population characteristics | No human research participants were used in this study. |
| Recruitment | No human research participants were used in this study. |
| Ethics oversight | No human research participants were used in this study. |

Note that full information on the approval of the study protocol must also be provided in the manuscript.

# Field-specific reporting

Please select the one below that is the best fit for your research. If you are not sure, read the appropriate sections before making your selection.

☒ Life sciences ☐ Behavioural & social sciences ☐ Ecological, evolutionary & environmental sciences

For a reference copy of the document with all sections, see nature.com/documents/nr-reporting-summary-flat.pdf

# Life sciences study design

All studies must disclose on these points even when the disclosure is negative.

| | |
|---|---|
| Sample size | No statistical methods were used to predetermine sample size. Sample sizes were determined based on previously published experiments. |
| Data exclusions | No data was excluded. |
| Replication | All experiments were performed in at least 3 biologically independent replicates with similar results. The experiments were performed independently and were reproducible. |
| Randomization | Randomization is not applicable as there are no groups to be allocated. |
| Blinding | Blinding was not applicable in this study as there are no groups to be allocated. Bias from prior knowledge described in the study is considered limited. |

# Reporting for specific materials, systems and methods

We require information from authors about some types of materials, experimental systems and methods used in many studies. Here, indicate whether each material, system or method listed is relevant to your study. If you are not sure if a list item applies to your research, read the appropriate section before selecting a response.

## Materials & experimental systems

| n/a | Involved in the study |
|---|---|
| ☐ | ☒ Antibodies |
| ☒ | ☐ Eukaryotic cell lines |
| ☒ | ☐ Palaeontology and archaeology |
| ☒ | ☐ Animals and other organisms |
| ☒ | ☐ Clinical data |
| ☒ | ☐ Dual use research of concern |
| ☒ | ☐ Plants |

## Methods

| n/a | Involved in the study |
|---|---|
| ☐ | ☒ ChIP-seq |
| ☐ | ☒ Flow cytometry |
| ☒ | ☐ MRI-based neuroimaging |

# Antibodies

| | |
|---|---|
| Antibodies used | E. coli RNA Polymerase beta Monoclonal Antibody clone 8RB13 mouse mAB (663905; Biolegend), Lex A Antibody (E-7) (sc-365999; Santa Cruz), 6X-His-tag Monoclonal Antibody (66005-1-Ig; Proteintech). |
| Validation | Antibodies were validated using western blot following manufacturer protocols with details provided in the Methods section. Antibodies used for ChIP were validated using western blot prior to performing experiments following manufacturer protocols. All antibodies have been used and cited extensively in house and in the literature. |

# Plants

| | |
|---|---|
| Seed stocks | N/A |
| Novel plant genotypes | N/A |
| Authentication | N/A |

# ChIP-seq

## Data deposition

☒ Confirm that both raw and final processed data have been deposited in a public database such as GEO.

☒ Confirm that you have deposited or provided access to graph files (e.g. BED files) for the called peaks.

| | |
|---|---|
| Data access links<br>*May remain private before publication.* | Sequencing data is available through the NCBI sequence read archive (SRA) under PRJNA992395 |
| Files in database submission | CHIP_WT_input_rep1.R1.fastq.gz<br>CHIP_WT_input_rep1.R2.fastq.gz<br>CHIP_WT_input_rep2.R1.fastq.gz<br>CHIP_WT_input_rep2.R2.fastq.gz<br>CHIP_WT_rep1.R1.fastq.gz<br>CHIP_WT_rep1.R2.fastq.gz<br>CHIP_WT_rep2.R1.fastq.gz<br>CHIP_WT_rep2.R2.fastq.gz<br>CHIP_25D_input_rep1.R1.fastq.gz<br>CHIP_25D_input_rep1.R2.fastq.gz<br>CHIP_25D_input_rep2.R1.fastq.gz<br>CHIP_25D_input_rep2.R2.fastq.gz<br>CHIP_25D_rep1.R1.fastq.gz<br>CHIP_25D_rep1.R2.fastq.gz<br>CHIP_25D_rep2.R1.fastq.gz<br>CHIP_25D_rep2.R2.fastq.gz |
| Genome browser session<br>(e.g. UCSC) | N/A |

## Methodology

| | |
|---|---|
| Replicates | ChIP-seq experiments were provided in biologically independent duplicate with reproducible results as seen in Extended Data Fig. 4. |
| Sequencing depth | We aimed for >10M paired-end reads per sample. |
| Antibodies | 6X-His-tag Monoclonal Antibody (66005-1-Ig; Proteintech). |

| Peak calling parameters | No peak calling was performed. Reads were counted within regions of interest around transcription start sites using Deeptools. See methods for more details. |
|---|---|
| Data quality | Data was reproducible between biologically independent replicates. More reads were found around transcription start sites as previously reported. |
| Software | CutAdapt (vl.18), Bowtie2 (v2.4.1), and deeptools2 (v3.5.l) |

# Flow Cytometry

## Plots

Confirm that:

☒ The axis labels state the marker and fluorochrome used (e.g. CD4-FITC).

☒ The axis scales are clearly visible. Include numbers along axes only for bottom left plot of group (a 'group' is an analysis of identical markers).

☒ All plots are contour plots with outliers or pseudocolor plots.

☒ A numerical value for number of cells or percentage (with statistics) is provided.

## Methodology

| Sample preparation | For DNA fragmentation experiments, cells were diluted 1,000-fold from overnight cultures and grown to an OD600 of 0.2 in 2mL of media before the addition of 80ug/mL of Rif . After 1 hour of treatment, cells were washed twice with an equal volume of LB before being fixed with 4% formaldehyde for terminal deoxynucleotide transferase dUTP nick end labeling (TUNEL). Labeling of DNA fragments was performed using the Apo-Direct Kit (BD Bioscience) following manufacturer instructions. |
|---|---|
| Instrument | FACSCalibur (BD Biosciences) |
| Software | FlowJo (vl0.6.0) |
| Cell population abundance | At least 50,000 cells were collected for each experimental condition. |
| Gating strategy | The percentage of TUNEL positive cells for a given condition is the percentage of cells exceeding the signal detected in >99% of untreated cells. |

☒ Tick this box to confirm that a figure exemplifying the gating strategy is provided in the Supplementary Information.

