## [Peer Review File · Nature]

Manuscript Title: High resolution landscape of an antibiotic binding site

Reviewer Comments & Author Rebuttals

Reviewer Reports on the Initial Version:

Referee expertise:

Referee #1: mechanisms of antibiotic action and resistance, MAGE

Referee #2: antibiotic resistance, RNAP and transcription-translation coupling

Referee #3: MAGE, synthetic biology, transcription

Referees' comments:

Referee #1 (Remarks to the Author):

The manuscript by Yang and colleagues describes the results of systematic mutagenesis of the rifampicin binding site in *E. coli* RNA polymerase (RNAP). Using MAGE, the authors generated a library of each possible single residue substitution at each position of the binding site of the protein. This library was then subject to pool-based screenings to determine the effect of these single substitutions both on fitness and on resistance to different antibiotics. Importantly, this screen identified not only more resistant mutants (which are easy to find also by selection), but also sensitivity-conferring mutations, which leads to reduce resistance compared to the wildtype. Interestingly, in these more sensitive mutants, the antibiotic had a bactericidal effect rather than the bacteriostatic effect that it had on the wildtype. Other interesting findings include mutations that affect RNAP rate and temperature activity range.

These results are novel and especially the transition from static to cidal activity is very interesting.

However, there are several issues with experimental design and analysis that I am concerned about:

Major comments:

1. Missing raw data precludes verifying the key results. The original data of T0 and T1 prior to normalization is missing. Only the ratio of T1 to T0 and of T1_antibiotic to T1_no-antibiotic is given. It appears that Fig. 1c provides an example of some of this un-normalized data, but it lacks the order of the amino acids in the x-axis, the position of the stop codon is unclear (is it the right-most column?), and it does not separate the data by each of the 64 codons. All raw data before normalization and before aggregating same-amino-acid codons should be provided in order to make an assessment of the analysis.

2. The sensitizing mutations appearing in fig 2a do not seem to appear in fig 1c. Comparing the rif column in Fig. 2a with the data in Fig. 1c (which in our understanding is the same data prior to normalization?), we do not see a signal of the hypersensitive mutants in Fig. 1c. Perhaps 1c is with no drug? The caption is not clear but seems to suggest that fig 1c T1 is after drug treatment.

3. The effect of different codons for the same amino acid is not analyzed.

The authors do not comment on the differences between different codons representing the same amino acids and do not provide the raw data. Comparing same-amino-acid codons can, at the very least, help shed light on the reliability and noisiness of the data.

4. MOST IMPORTANTLY: The mutant response to antibiotics should be compared at the same fold-change above their respective MIC, not at the same absolute concentrations. The main finding of the paper, namely the mutations that change from static to cidal activity, rests on a misleading comparison of mutant response to the drug at the same absolute concentrations (Fig. 3e). However, these mutants have substantially different MICs and the entire difference we see in drug effect could simply reflect the MIC difference. The experiments of viability (to detect cidal vs static) should be repeated with the different strains compared at the same fold-change above MIC, not at the same absolute concentrations. Even better, it will be illuminating to compare the viability of the mutants as a function of MIC-normalized dosage (an MIC-normalized dose-response curve).

Minor comments:

The statistical analysis is not well explained. How were Z-scores determined? How are p-values calculated? Do they reflect the variance in the replicates? Or the read Poisson statistics? Something else?

Figure 3a shows the same data shown as one column in Figure 2a, repetitive presentation of the same data is not recommended.

In Fig. 2a it is not explicitly clear whether red indicates sensitive or resistant mutants.

Between Figure 3a and 3b the overall sign of the values seems to change (positive fold-change values are suddenly negative and vice versa).

Referee #2 (Remarks to the Author):

Manuscript by Yang et al described the generation and characterization of a saturated library of RNA polymerase mutants containing almost all possible mutations at the rifampicin binding site and showed a higher-than-expected mutability of the rifampicin binding pocket region. By measuring the

growth fitness and susceptibility to the bacteriostatic antibiotic rifampicin (RIF), transcription termination inhibitor bicyclomycin (BCM), and nucleotide analogue 5FU, the study revealed diverse phenotypes associated with the mutation signatures in the RNAP. First, the study identified new RNAP mutations that cause hypersensitivity to RIF. Further investigation of these hypersensitive mutants led to an interesting hypothesis that increasing the affinity of RNAP to RIF can render RIF treatment from bacteriostatic to bactericidal. The mechanism of RIF lethality appears to involve the generation of lethal DNA damage and fragmentation rather than presumed inhibition of transcription. Second, by examining the susceptibility of RNAP mutants to BCM and 5FU, the study proposed that RNAPs with higher elongation rate are more sensitive to BCM and 5FU due to rapid depletion of pyrimidine nucleotides, while slower RNAPs are more resistant. This suggests that alteration of RNAP properties can lead to unexpected response to agents that does not directly target transcription. Overall, the study started with a novel and elegant design, presented a wealth of interesting observations and novel hypothesis regarding RNAP properties and antibiotic actions. The current manuscript provided only limited evidence to support these intriguingly hypotheses. The authors should follow through their models by providing additional mechanistic evidence to support causality and generality of their conclusions.

Major comments:

1. The authors claimed that Rif kills susceptible *rpoB* mutants by inducing DNA damage due to replication transcription conflicts. The association of DNA damage and Rif lethality is interesting but could simply be a symptom of cell death rather than a primary cause of death. Is it possible to monitor Rif-mediated death in the mutant to differentiate the sequence of events? The authors also argue that this DNA damage arise from transcription-replication conflicts due to prolonged Rif binding to the transcription complexes, which need to be supported by experimental evidence. The alternative hypothesis that Rif kills *rpoB* mutants by generating excessive ROS, which directly damage DNA without conflicts, should be considered.
 - a. Rif treatment of WT and Rif hypersensitive mutant both lead to strong accumulation of 3-mer indicative of RNAP obstruction, suggesting DNA damage occur in both cases but only the mutant died. TUNEL assay showed a fraction of cells having significant accumulation of DNA double strand break (WT ~15%; Rif-sensitive 25-35%) after Rif treatment. However, according to the plating efficiency assay, WT weren't killed at all, even if there are 15% cells showing high level of DNA double strand break; Rif-sensitive strains got over 90% cells killed, while only 25-35% of the cells showed high level of DNA double strand break. How to explain this discrepancy?
 - b. The SOS response in Rif-sensitive mutants was quantified by RT-qPCR. Given that Rif itself is strongly inhibiting transcription, the normalization standard *rpsL*'s transcription will be strongly inhibited compared to its normally high transcription, which in return may lead to over-estimate of *umuC* and *recA* transcription. Can the authors provide an alternative method than RT-qPCR to show that SOS is induced? (i.e by LexA cleavage or RNA-seq).
 - c. Can the authors elaborate their model for replisome collisions with transcription complex? Backed up array of RNA polymerase impedes replication. R-loop also causes conflicts. However, Rif stops RNA polymerase at 3nt into the promoter and there will be no backed up array or R-loop. Do the authors suggest that it is the transcription initiation complex that blocks the replisome directly with Rif binding increasing the affinity of this complex to the promoter?
2. The authors claimed that Rif-resistant mutants with faster transcription speed will deplete pyrimidine nucleotides to lead to 5FU susceptibility. However, the alternative hypothesis is that *rpoB*

mutants globally reprogrammed transcription initiation to change expression levels of genes involved in pyrimidine synthesis and consumption. The authors should provide evidence such as RNA-seq to rule it out.

a. The authors detected lowered pyrimidine nucleotide levels in Rif-resistant mutants. UTP and CTP quantified in this experiment cannot represent dTMP level, which is directly competing with the 5FU incorporation. Given that UTP consumption and dTMP synthesis pathways are both downstream of UDP synthesis, it is hard to believe that over consumption of UTP itself will lead to lowered dTMP and thus higher chance of 5FU incorporation.

b. The degrees of transcription elongation speed do not correlate well with 5FU sensitivity. 15Y and 26V mutants have similar transcription speed, and 15Y has transcription pause frequency even closer to WT than 26V, which indicates that 26V may consume NTPs even faster than 15Y. However, the UTP level and CTP level of 15Y is even lower than 26V, and 15Y is more susceptible to 5FU than 26V. How to explain this discrepancy?

other comments:

1. The study often used the term “MIC” to describe the susceptibility of the mutants, but the term in fact was defined as “Mean Inhibitory Concentration” determined by inhibition of growth below OD600 of 0.2. This is different from the conventional Minimal Inhibitory Concentration, also known as MIC, which is defined by the lowest drug concentration to achieve complete inhibition of growth. Is there a reason why this alternative assay is used over the conventional MIC determination?

2. Antibiotic activities are commonly defined by their MBC (minimal bactericidal concentration) which reflects bacterial killing, and MIC (minimal inhibitory concentration) which reflects growth inhibition. Generally, antibiotic with MBC/MIC ratios ≤ 4 are defined as bactericidal and ratios > 4 are bacteriostatic. Since recovery from lag can result from either or both changes in bacterial killing and inhibition, what are the MIC and MBC of the hypersensitive mutants to Rif compared to WT? This would help to differentiate specific defects of RNAP mutations.

3. Figure 5c: the legend seems to be not corresponding to the figure. It claims to be “heatmap of rpoB mutant fitness and sensitivity or resistance to rifampicin, bicyclomycin, and 5-fluorouracil”, but the actual figure is a heatmap comparing the frequencies of mutations among different bacterial species.

4. It is mentioned that Rif is bactericidal to gram positive bacteria but bacteriostatic to E. coli, can this difference be attributed to different affinity of their RNAPs to Rif? Also, does Rif lethality in those organisms involve DNA damage? This can enhance generality of the model.

Referee #3 (Remarks to the Author):

The manuscript by Yang et al describes the generation of saturation mutagenesis mutants in the Rif binding site of RNAP and its mechanistic characterizations. The authors identified an alpha helix whose mutations was observed to have high sensitivity to Rif. Single residue mutants that increased the toxicity of Rif to a level beyond bacteriostatic to a bactericidal drug was also seen. Interestingly, the authors found mechanism of killing to be associated with DNA damage. Mutations that increased RNAP speed also showed increased sensitivity to other drugs and growth advantage in temperature shifts. The authors also performed some evolutionary analysis to look at the distributions of natural mutations in the Rif binding sites of RNAP in other bacteria.

Overall, this work appears to be done rigorously. The experiments and data presented are compelling and generally support the conclusions of the work. The work is pretty comprehensive in delineating the effects and consequences of these Rif binding site mutations. In terms of significance, there's some questions of whether the bactericidal effects of these mutations are clinically relevant given that they are generally not enriched under any natural situation. Recently another study identified compensatory mutations in RNAP under Rif resistance that increase RNAP rates (Doi: 10.1093/nar/gkac406), which is consistent with data shown in this manuscript. The work is a nice example of systematically generating mutants for more detailed mechanistic studies in understanding the multi-dimensional fitness landscape of antibiotic resistance mutants. How these insights convert to better and more potent antimicrobial compounds remains to be seen.

Some additional comments:

- Can the authors discuss why some of the STOP codon mutations are enriched or not highly selected against? (e.g. position 526 in Fig 3, positions 513, 517 in Extended Data Fig1, or positions 521, 569, 570, 572 in Extended Data Fig2)
- Do all fast RNAP mutants deplete nucleotides? The authors characterized a handful of mutants for characterization and the data is compelling overall, but I wonder if this is true for all observed mutants. Are there mutations that can be introduced in RNAP to slow it down to show that this is the major mechanism and not from other errant aspects of transcriptional changes?
- Authors noted that the fast RNAP mutants are more fit in 30C vs WT where the screen was done but not at 37C. What conditions were the 5FU and BCM done? The authors showed data for 25C and 42C for 5FU, but unclear it is unclear about 30C or 37C. Did the authors do the Rif screen at 37C, which is probably more clinically relevant?
- Figure 4a, please show fit/correlation values with P-values.
- The authors state in the discussion that "Rif kills cells through DNA damage and not transcription inhibition per se." The study shows that the in hyper-sensitive Rif mutants, there is an additional component of DNA damage involved that yields more bactericidal effects. There should be more nuanced discussion of such points.
- In general, it's a little unclear whether the last analysis for Rif binding mutations in nature yielded any novel insights. Some of the results about conservation is not surprising. The observation of 24L enriched in species, which is a fast RNAP mutant in E. coli, is interesting. However, one cannot know for sure that such mutation also yields the fast RNAP phenotype in the rest of RNAP genetic background in those organisms so the results are speculative at best.

Author Rebuttals to Initial Comments:

REFEREE #1

The manuscript by Yang and colleagues describes the results of systematic mutagenesis of the rifampicin binding site in *E. coli* RNA polymerase (RNAP). Using MAGE, the authors generated a library of each possible single residue substitution at each position of the binding site of the protein. This library was then subject to pool-based screenings to determine the effect of these single substitutions both on fitness and on resistance to different antibiotics. Importantly, this screen identified not only more resistant mutants (which are easy to find also by selection), but also sensitivity-conferring mutations, which leads to reduce resistance compared to the wildtype. Interestingly, in these more sensitive mutants, the antibiotic had a bactericidal effect rather than the bacteriostatic effect that it had on the wildtype. Other interesting findings include mutations that affect RNAP rate and temperature activity range.

These results are novel and especially the transition from static to cidal activity is very interesting.

However, there are several issues with experimental design and analysis that I am concerned about:

We thank the reviewer for their helpful comments and their enthusiasm for the novel findings of the manuscript. We have resolved the issues with experimental design and analysis as detailed below.

Major comments:

1. Missing raw data precludes verifying the key results. The original data of T0 and T1 prior to normalization is missing. Only the ratio of T1 to T0 and of T1_antibiotic to T1_no-antibiotic is given. It appears that Fig. 1c provides an example of some of this un-normalized data, but it lacks the order of the amino acids in the x-axis, the position of the stop codon is unclear (is it the right-most column?), and it does not separate the data by each of the 64 codons. All raw data before normalization and before aggregating same-amino-acid codons should be provided in order to make an assessment of the analysis.

We have now added additional supplementary figures displaying all raw data prior to normalization (**new Extended Data Fig. 1**). Each of the MAGE-seq experiments were conducted in triplicate, and we present in the new figures the mean for each condition. We note that each drug treated condition was normalized for fitness, and not to T0. For example, rif_mean (**new Extended Data Fig. 1b**) is not normalized to T0 for presentation in Fig. 3a. It is normalized for fitness, similar to the data presented in t1_mean (**new Extended Data Fig. 1a**).

The reviewer is correct that Fig. 1c is a representative figure of mutant abundances at T0 compared to T1. Fig. 1 is meant to be a schematic of the MAGE screening performed. We include Fig 1c to help readers understand the normalization being performed and made this more clear in the figure legend. We have also added the amino acid and position labels as requested (**revised Fig. 1c**).

We apologize for being unclear about the design of MAGE-seq oligos. As mentioned in the methods section, we designed specific codons to produce each desired amino acid substitution. We expand on our reasoning for this below in response to comment 3.

2. The sensitizing mutations appearing in fig 2a do not seem to appear in fig 1c. Comparing the rif column in Fig. 2a with the data in Fig. 1c (which in our understanding is the same data prior to normalization?), we do not see a signal of the hypersensitive mutants in Fig. 1c. Perhaps 1c is with no drug? The caption is not clear but seems to suggest that fig 1c T1 is after drug treatment.

Fig. 1c is a representative figure meant to help general readers understand the normalization performed, as mentioned in the legend. We have improved the description in the legend. To improve clarity, we have replaced the second panel of Fig. 1c with T1 fitness, with no drug treatment, and provide more detailed labeling (**revised Fig. 1c**). We also display heatmaps of all raw data prior to normalization in the supplement (**new Extended Data Fig. 1**).

3. The effect of different codons for the same amino acid is not analyzed. The authors do not comment on the differences between different codons representing the same amino acids and do not provide the raw data. Comparing same-amino-acid codons can, at the very least, help shed light on the reliability and noisiness of the data.

For MAGE recombineering, we designed specific oligos that produced the desired single amino acid substitutions with multiple nucleotide alterations when possible, as described in the methods. We did this to limit the effect of background mutations, to improve the uniformity of mutants in the pool, and to increase the chances of detecting all of the possible mutants generated. The ability to detect all mutations was particularly important because we were interested in mutant depletion after treatment (i.e. to detect drug sensitivity). It is unlikely we would have been able to achieve such uniform mutagenesis (as seen in Fig. 1c and Extended Data Fig. 1) if we used random nucleotide sequences such as 'NNN' or 'NNK' for recombineering, since they disproportionately represent certain amino acids.

Our work was primarily focused on the effects of amino acid substitutions. However, the effects of same-amino-acid codons is also very interesting. Elegant and rigorous work utilizing such random nucleotide sequences to analyze the effects of codon usage is cited in our paper (PMID: 28009265).

In regards to reliability and noisiness of the data, all MAGE screening experiments were done in triplicate, which allowed us to calculate statistics for each position as presented in the volcano plots in Extended Data Fig. 2-4, and Fig. 3b.

4. MOST IMPORTANTLY: The mutant response to antibiotics should be compared at the same fold-change above their respective MIC, not at the same absolute concentrations. The main finding of the paper, namely the mutations that change from static to cidal activity, rests on a misleading comparison of mutant response to the drug at the same absolute concentrations (Fig. 3e). However, these mutants have substantially different MICs and the entire difference we see in drug effect could simply reflect the MIC difference. The experiments of viability (to detect cidal vs static) should be repeated with the different strains compared at the same fold-change above MIC, not at the same absolute concentrations. Even better, it will be illuminating to

compare the viability of the mutants as a function of MIC-normalized dosage (an MIC-normalized dose-response curve).

This is a great point. We repeated the same killing experiment with 4x and 10x MIC normalized dosage and measured plating efficiency of hypersensitive strains. We observed a similar magnitude of killing at both concentrations (**new Extended Data Fig. 6b**).

We also determined the MIC and MBC of rifampicin adhering to established clinical guidelines by CLSI and EUCAST (PMID:18274517). While the prior killing assay is done in exponentially growing cells in regular LB broth with 1 hour of Rif treatment at 37C, these assays are done in Mueller-Hinton (MH) broth with a standardized inoculum for 18 hours at 37C. Again, we see a mild decrease in the MIC of hypersensitive mutants in MH broth (**new Extended Data Fig. 6c**). In addition, we determine the MBC of hypersensitive strains to be close to 2x their respective MIC (**new Extended Data Fig. 6d**). Rif remains bacteriostatic at the highest concentration tested in wild-type cells (80ug/mL) (**new Extended Data Fig. 6d**). As a result, the MBC/MIC ratio of hypersensitive strains is ≤ 4 (**new Extended Data Fig. 6e**). These results demonstrate that Rif is bactericidal in hypersensitive strains across different experimental conditions.

Minor comments:

The statistical analysis is not well explained. How were Z-scores determined? How are p-values calculated? Do they reflect the variance in the replicates? Or the read Poisson statistics? Something else?

We apologize for being unclear. The Z-scores in Fig. 2a are calculated per condition (columns) for visualization purposes, in order to compare the effects of each drug on each mutant. This is typically done when presenting data in a heatmap (such as for RNAseq) to prevent large changes in one condition from overwhelming patterns or trends in a different condition. For example, the fold changes in the high bicyclomycin condition are much larger when compared to the low concentration. If both datasets are presented unnormalized in the same heatmap, it might be difficult to identify patterns and trends in the low bicyclomycin dataset. We now provide these details in the figure legend.

Other data presentations of MAGE screening results (such as in Fig. 3a, Extended Data Fig. 2-4) are presented as log₂ fold changes between T1 and T0. As all MAGE screening experiments were performed in triplicate, P values were calculated with the independent t-test (two-sided, equal variance) and reflect variance between replicates. We have included these details in the legend and methods.

Figure 3a shows the same data shown as one column in Figure 2a, repetitive presentation of the same data is not recommended.

We agree that repetition is undesirable, however this presentation of the data is intentional as it provides readers with different visualizations of the same dataset.

We present the data normalized by Z-score in Fig. 2a for visualization purposes so that readers can compare the effects of each drug on each mutant. In Fig. 3a, the data is presented as log₂ fold change to allow readers to appreciate the relative change in mutant abundance between T1

and T0. In addition, the data in Fig. 3a is provided in a way that allows readers to observe both the position and residue-specific effects of mutations; in Fig. 2a only the positional effects can be observed. For example, in Fig. 3a can it be readily appreciated that Rif hypersensitivity occurs with a single mutation at position 521 (L->Y), while multiple substitutions produce the phenotype at position 525.

We provide similar presentations of the other 5 screening conditions presented in Fig. 2a in the extended data. If the reviewer still believes Fig. 3a is redundant, we can move it to the extended data as well.

In Fig. 2a it is not explicitly clear whether red indicates sensitive or resistant mutants.

We have modified the figure and legend to make it clear that red indicates resistant mutants (**revised Fig. 2a, 3a**).

Between Figure 3a and 3b the overall sign of the values seems to change (positive fold-change values are suddenly negative and vice versa).

This misunderstanding might be improved with labeling resistant mutants as red and sensitive mutants as blue as with the prior comment. Fig. 3a and 3b should be consistent. In Fig. 3b, we highlight 21Y and 25D as the most sensitive mutants, with a negative log₂ fold change. In Fig. 3a, these mutants are also dark blue, with a negative log₂ fold change.

REFEREE #2

Manuscript by Yang et al described the generation and characterization of a saturated library of RNA polymerase mutants containing almost all possible mutations at the rifampicin binding site and showed a higher-than-expected mutability of the rifampicin binding pocket region. By measuring the growth fitness and susceptibility to the bacteriostatic antibiotic rifampicin (RIF), transcription termination inhibitor bicyclomycin (BCM), and nucleotide analogue 5FU, the study revealed diverse phenotypes associated with the mutation signatures in the RNAP. First, the study identified new RNAP mutations that cause hypersensitivity to RIF. Further investigation of these hypersensitive mutants led to an interesting hypothesis that increasing the affinity of RNAP to RIF can render RIF treatment from bacteriostatic to bactericidal. The mechanism of RIF lethality appears to involve the generation of lethal DNA damage and fragmentation rather than presumed inhibition of transcription. Second, by examining the susceptibility of RNAP mutants to BCM and 5FU, the study proposed that RNAPs with higher elongation rate are more sensitive to BCM and 5FU due to rapid depletion of pyrimidine nucleotides, while slower RNAPs are more resistant. This suggests that alteration of RNAP properties can lead to unexpected response to agents that does not directly target transcription. Overall, the study started with a novel and elegant design, presented a wealth of interesting observations and novel hypothesis regarding RNAP properties and antibiotic actions. The current manuscript provided only limited evidence to support these intriguingly hypotheses. The authors should follow through their models by providing additional mechanistic evidence to support causality and generality of their conclusions.

We thank this reviewer for their careful reading of our work and their acknowledgement of the wealth of interesting observations and novel hypothesis generated in this work. We now provide substantial additional data in support of the models proposed.

Major comments:

1. The authors claimed that Rif kills susceptible *rpoB* mutants by inducing DNA damage due to replication transcription conflicts. The association of DNA damage and Rif lethality is interesting but could simply be a symptom of cell death rather than a primary cause of death. Is it possible to monitor Rif-mediated death in the mutant to differentiate the sequence of events?

This is an excellent point. The reviewer first asks if DNA damage by Rif is a symptom rather than the primary cause of cell death. We find this to be unlikely for two reasons: cells deficient in DNA repair are further sensitized to killing, and cells induce the SOS response in response to Rif (expanded on in major comment 1b). We highlight that hypersensitive mutants lacking DNA repair proteins *recA* or *recBCD* are further sensitized to killing by Rif (Fig. 3k). Moreover, wild-type cells deleted for *recA* are killed by Rif (Fig. 3k). Evidently, a functional DNA repair system protects cells from the bactericidal effects of Rif, indicating that DNA damage is not just an inconsequential byproduct of cell death. Induction of the SOS response in response to Rif also provides further evidence that cells must react to Rif-induced DNA damage in order to survive (**new Extended Data Fig. 8**, expanded on in major comment 1b).

The authors also argue that this DNA damage arise from transcription-replication conflicts due to prolonged Rif binding to the transcription complexes, which need to be supported by experimental evidence. The alternative hypothesis that Rif kills *rpoB* mutants by generating excessive ROS, which directly damage DNA without conflicts, should be considered.

In the original version of the manuscript, transcription-replication conflicts only appeared as speculation in the discussion. As requested by the reviewer, we have greatly expanded upon this model in response to major comment 1c.

The reviewer also suggests exploration of an alternative model where excessive ROS directly damages DNA. This is likely in reference to comprehensive work by the Collins group describing increased cellular respiration and buildup of toxic ROS as a common mechanism of cellular death induced by bactericidal antibiotics (PMID: 17803904). In this seminal work they saw no ROS increase in cells treated with bacteriostatic antibiotics, including Rif. Moreover, follow up work from the Collins lab studying the bactericidal effects of Rif in *S. aureus* found no increase in cellular respiration (PMID: 26100898), suggesting that the mechanism of killing by Rif is distinct from other bactericidal antibiotics.

To test if the bactericidal effect in hypersensitive mutants is associated with an increase in ROS, we measured fluorescence of the ROS indicator carboxy-H2DCFDA before and after ampicillin and Rif treatment, under the same conditions we observe killing. As previously reported, treatment of cells with the bactericidal antibiotic ampicillin greatly increases ROS, while treatment with Rif does not (**new Extended Data Fig. 9a**). In fact, Rif treatment decreases ROS, in agreement with previous observations that bacteriostatic drugs generally decrease cellular respiration (PMID: 26100898).

Importantly, killing by bactericidal antibiotics was shown to be rescued with the iron chelator 2,2'-dipyridyl and the hydroxyl radical quencher thiourea. We conducted the same experiments and show that while ampicillin killing can be rescued with these compounds, Rif killing cannot (**new Extended Data Fig. 9b, c**). Together these results suggest that excessive ROS does not contribute to the bactericidal effect of Rif in hypersensitive mutants.

a. Rif treatment of WT and Rif hypersensitive mutant both lead to strong accumulation of 3-mer indicative of RNAP obstruction, suggesting DNA damage occur in both cases but only the mutant died. TUNEL assay showed a fraction of cells having significant accumulation of DNA double strand break (WT ~15%; Rif-sensitive 25-35%) after Rif treatment. However, according to the plating efficiency assay, WT weren't killed at all, even if there are 15% cells showing high level of DNA double strand break; Rif-sensitive strains got over 90% cells killed, while only 25-35% of the cells showed high level of DNA double strand break. How to explain this discrepancy?

This is an important observation and appears to us as a threshold effect where cells can tolerate and repair a certain amount of DNA damage before succumbing to cell death. Other bactericidal drugs have been shown to cause variable amounts of DNA damage that do not correlate with the magnitude of loss in viability. For example, the percentage of TUNEL positive *E. coli* cells after 1.5 hours treatment with 750ng/mL norfloxacin is less than 30%, however more than 99.9% of cells are killed (PMID: 22633370). Another potential confounding factor is that the TUNEL assay is only able to detect DNA damage in intact cells that have not completely lysed.

This interpretation is also consistent with our finding that *recA* wild-type *rpoB* cells also become sensitive to Rif killing due to impaired DNA repair (Fig. 3k). The observation that *recBCD* or *recA* deletion in hypersensitive mutants increases Rif sensitivity also indicates that functional DNA repair is crucial for survival in response to Rif treatment (Fig. 3k). Importantly, we verified that these effects are specific to Rif, since other bacteriostatic drugs have no effect on *recA* or *recBCD* cells (**new Extended Data Fig. 7b**).

To provide an additional indication of DNA damage, we designed a semi-long range qPCR assay (PMID: 19966269) to detect DNA breaks at the promoters and over the gene body of two highly expressed genes after Rif treatment (**new Extended Data Fig. 10a, b**). Interestingly, we observed a significant increase in DSB at gene promoters but not gene bodies (**new Fig. 3j**), providing additional evidence that Rif causes DNA damage.

b. The SOS response in Rif-sensitive mutants was quantified by RT-qPCR. Given that Rif itself is strongly inhibiting transcription, the normalization standard *rpsL*'s transcription will be strongly inhibited compared to its normally high transcription, which in return may lead to over-estimate of *umuC* and *recA* transcription. Can the authors provide an alternative method than RT-qPCR to show that SOS is induced? (i.e by LexA cleavage or RNA-seq).

The reviewer is correct that transcription inhibition by Rif may complicate the interpretation of RT-qPCR assays. To provide an additional indicator of SOS induction not dependent on transcription, we measured *lexA* protein levels before and after Rif treatment by western blot. We observe that *lexA* protein decreases in Rif treated cells, indicating that *lexA* is less available to function as a repressor (**new Extended Data Fig. 8b, c**). As a control, we measured *rpoB*

protein levels in the same samples and observed that they remain consistent before and after Rif treatment (**new Extended Data Fig. 8b, c**). This suggests that transcription inhibition by Rif is not responsible for the decrease in *lexA* protein. These results provide additional evidence that SOS is induced in response to Rif treatment.

c. Can the authors elaborate their model for replisome collisions with transcription complex? Backed up array of RNA polymerase impedes replication. R-loop also causes conflicts. However, Rif stops RNA polymerase at 3nt into the promoter and there will be no backed up array or R-loop. Do the authors suggest that it is the transcription initiation complex that blocks the replisome directly with Rif binding increasing the affinity of this complex to the promoter?

In the original version of the manuscript, transcription-replication conflicts was one possible mechanism for DNA damage speculated on in the discussion. We now have greatly expanded on this model as requested by the reviewer.

Since we show ROS is unlikely to be involved (major comment 1, **new Extended Data Fig. 9**), we considered other possible sources for DNA damage. Prior work has shown that promoter-arrested RNAP presents an obstacle for the replisome, resulting in interference with replication fork progression or deleterious mutations at the promoter (PMID: 27362223, 16670199). These mutations at the promoter were dependent on *recA*, indicating that they are caused by recombination after replication fork collision (PMID: 27362223). This is consistent with our data that shows without *recA*, Rif becomes potently bactericidal in both wild-type and hypersensitive mutants (**Fig. 3k**).

Because we also observed that DNA damage appears to occur specifically at promoters (major comment 1a, **new Fig. 3j**), we propose that replication-transcription conflicts contribute to DNA damage by Rif. The reviewer is correct that in this model, Rif blocking the transcription initiation complex increases the likelihood of collisions with the replisome. We show *in vitro* that hypersensitive mutant RNAP remains bound to the promoter longer after Rif binding (**Fig. 3f**). We now also provide ChIP-seq data from wild-type and hypersensitive RNAP after recovery from Rif treatment and observe that more hypersensitive RNAP remains bound to the promoter *in vivo* (**new Fig. 3g, new Extended Data Fig. 5a**).

To determine if replication conflicts at promoters are involved in DNA damage by Rif, we introduced dCas9 targeting the *oriC* to hypersensitive strains to block replication prior to Rif treatment (**new Extended Data Fig. 10c**). We found that blocking replication significantly decreases the frequency of DSB breaks at promoters (**new Extended Data Fig. 10d, e**).

Next, we constructed hypersensitive mutants in cells containing a temperature-sensitive allele of the replication protein *dnaC2* (PC2) (PMID: 4925091, 20478253). We verified that shifting cells to the non-permissive temperature 42C inhibits replication (**new Extended Data Fig. 10f**). We then compared the bactericidal effects of Rif with and without replication. Strikingly, we observe that Rif is no longer bactericidal in hypersensitive mutants when replication is inhibited (**new Fig. 3l**). Rif remains bactericidal for hypersensitive mutants in the PC2 background at the permissive temperature of 30C (**new Extended Data Fig. 10g**).

In summary, we show that Rif induces DNA damage at promoters, and the bactericidal effects and DNA damage by Rif are dependent on replication. This data, combined with previous work characterizing replisome conflicts with promoter-bound RNAP, provides evidence that Rif damages DNA through transcription-replication conflicts.

2. The authors claimed that Rif-resistant mutants with faster transcription speed will deplete pyrimidine nucleotides to lead to 5FU susceptibility. However, the alternative hypothesis is that *rpoB* mutants globally reprogrammed transcription initiation to change expression levels of genes involved in pyrimidine synthesis and consumption. The authors should provide evidence such as RNA-seq to rule it out.

We have completed RNA-seq of the fast mutants 15Y and 27W and observe that no major transcriptional reprogramming occurs in genes involved in pyrimidine ribonucleotide biosynthesis and consumption. The only significant changes (\log_2 fold change >2) are in the expression of *pyrD*, a gene encoding dihydroorotate dehydrogenase, involved in de novo pyrimidine ribonucleotide biosynthesis and in *upp*, a gene encoding uracil phosphoribosyltransferase, involved in pyrimidine ribonucleotide salvage (**new Extended Data Fig. 14a**).

To test if decreased levels of *pyrD* is responsible for differential 5FU sensitivity of fast mutants, we reconstructed fast mutants in a *pyrD* background and observed no rescue of 5FU sensitivity (**new Extended Data Fig. 14b**). *pyrD* deletion in wild-type cells also had little effect on 5FU sensitivity (**new Extended Data Fig. 14b**, Fig. 4g).

Upp downregulation in fast mutants is a more surprising finding since pyrimidine ribonucleotide salvage is also responsible for activation of 5FU. One would expect *upp* downregulation to increase resistance to 5FU, and this is indeed what we observe in wild-type cells and fast mutants lacking *upp* (**new Extended Data Fig. 14c**, Fig. 4g). However, *upp* fast mutants are still hypersensitive to 5FU when compared to *upp* wild-type cells (**new Extended Data Fig. 14c**). We conclude that transcriptional reprogramming is unlikely to be a primary mechanism for 5FU sensitivity in fast *rpoB* mutants.

The mechanism for differential regulation of *upp* and *pyrD* may be due to an additional, non-consequential attribute of fast mutants. The *carAB*, *pyrBI*, *upp-uraA*, and *codAB* operons are known to be regulated by a UTP-sensing mechanism of slippage and reiterative transcription while *pyrC* and *pyrD* are thought to respond to CTP by a different mechanism (see comprehensive review PMID: 18535147). The downregulation of *upp* and *pyrD* might suggest that fast mutants are more likely to slip and abort transcription of these genes. Alternatively, downregulation of *upp* might be a compensatory mechanism by cells to limit the conversion of imported uracil into uridine monophosphate, so that uracil can instead be used to produce deoxyuridine and then replenish dTMP through *deoA*, *tdk*, and *thyA*.

a. The authors detected lowered pyrimidine nucleotide levels in Rif-resistant mutants. UTP and CTP quantified in this experiment cannot represent dTMP level, which is directly competing with the 5FU incorporation. Given that UTP consumption and dTMP synthesis pathways are both downstream of UDP synthesis, it is hard to believe that over consumption of UTP itself will lead to lowered dTMP and thus higher chance of 5FU incorporation.

The reviewer asks for more evidence that higher UTP consumption leads to dTMP depletion in fast Rif binding site mutants. The reviewer is correct that both UTP and dTMP can be produced from UDP, and is skeptical that the higher consumption of UTP we observe (Fig. 4f) can make dTMP more scarce.

We provide two additional pieces of evidence that nucleotide pools and dTMP are depleted in fast *rpoB* mutants. First, we tested if fast mutants are more susceptible to thymineless death by reconstructing fast mutants 15Y, 26V, and 27W in a *thyA* background and starving them of thymidine. *thyA* mutants are unable to produce their own thymidine and must be supplemented with it to support growth. When thymidine is removed, cells are able to subsist on the existing reservoirs of dTMP during a “resistance phase” before viability is lost (PMID: 33318216). In support of our model of faster depletion of intracellular thymidine by fast RNAP, fast mutants in a *thyA* background lose viability more rapidly than wild-type *thyA* cells upon thymidine starvation (**new Fig. 4i**).

Next, we measured sensitivity of fast mutants 15Y, 26V, and 27W to trimethoprim. Trimethoprim inhibits dihydrofolate reductase, which is required for both purine and dTMP production. Consistent with our model for nucleotide and dTMP depletion, fast mutants are also sensitive to trimethoprim (**new Fig. 4h**). This demonstrates that nucleotide depletion by fast RNA polymerase mutants can also increase sensitivity to other clinically relevant antibiotics.

It is important to note that pyrimidine ribonucleotides are the primary source for synthesis of dTMP in media which is not supplemented with thymine or thymidine. The conditional essentiality of *thyA* KO in LB media indicates that LB does not provide enough thymine/thymidine for the dTMP salvage pathway to support growth. Assuming their synthesis capacity is not unlimited, over-consumption of pyrimidine ribonucleotides likely limits the availability of dUMP required for de novo synthesis of dTMP.

In summary, we show that fast mutants are: depleted for nucleotides using metabolomics (Fig. 4f); are more sensitive to the nucleotide analogue 5FU (Fig. 4g); can be rescued from 5FU sensitivity with thymidine supplementation (Fig. 4g); are more sensitive to trimethoprim, a clinically relevant antibiotic that kills cells by starving them of dTMP (**new Fig. 4h**); and are more sensitive to thymidine starvation in thymineless death assays (**new Fig. 4i**). These results are all in support of our model that fast mutants deplete nucleotides, altering 5FU sensitivity.

b. The degrees of transcription elongation speed do not correlate well with 5FU sensitivity. 15Y and 26V mutants have similar transcription speed, and 15Y has transcription pause frequency even closer to WT than 26V, which indicates that 26V may consume NTPs even faster than 15Y. However, the UTP level and CTP level of 15Y is even lower than 26V, and 15Y is more susceptible to 5FU than 26V. How to explain this discrepancy?

The 15Y mutant is particularly interesting because at 37C and 42C it suffers from a growth defect in LB broth in the absence of 5FU (**new Extended Data Fig. 15b**, 17a). This makes it challenging to work with for assays measuring transcription speed, NET-seq, or metabolomics. The discrepancy might arise because in transcription speed and NET-seq assays we collected cells at an OD of 0.4, while in metabolomic assays we collected cells after a fixed length of time for analysis. Furthermore, pause frequency might not correlate exactly with transcription speed,

since pause frequency is measured genome wide using NETseq and transcription speed is measured at a single locus (see Methods).

We believe that the severe depletion of nucleotides in this mutant might cause this growth defect (Fig. 4f). Interestingly, adding 0.1 or 1mg/mL thymidine to growth media completely rescues the growth defect of 15Y (**new Extended Data Fig. 15b**). This suggests that LB is lacking in thymidine, which is also why *thyA* mutants are unable to grow in LB without thymidine supplementation. This also provides further support for our observation of nucleotide depletion by fast mutants.

other comments:

1. The study often used the term 'MIC' to describe the susceptibility of the mutants, but the term in fact was defined as 'Mean Inhibitory Concentration' determined by inhibition of growth below OD600 of 0.2. This is different from the conventional Minimal Inhibitory Concentration, also known as MIC, which is defined by the lowest drug concentration to achieve complete inhibition of growth. Is there a reason why this alternative assay is used over the conventional MIC determination?

We apologize for the typo, minimal inhibitory concentration is correct. The value of OD600 of 0.2 did not account for background reading of LB blanks in our Bioscreen machine, which typically gave a reading of around 0.1. Subtracting the background reading gives a cutoff OD600 of 0.1 which would correspond with >90% growth inhibition. Guidelines for broth microdilution assays recommend reading MIC 'as the lowest concentration of antimicrobial agent that completely inhibits growth of the organism by the unaided eye' (PMID: 18274517). We used this cutoff of >90% in the analysis of MIC assays done in liquid format because it reflects the lowest OD we could visualize growth.

2. Antibiotic activities are commonly defined by their MBC (minimal bactericidal concentration) which reflects bacterial killing, and MIC (minimal inhibitory concentration) which reflects growth inhibition. Generally, antibiotic with MBC/MIC ratios ≤ 4 are defined as bactericidal and ratios > 4 are bacteriostatic. Since recovery from lag can result from either or both changes in bacterial killing and inhibition, what are the MIC and MBC of the hypersensitive mutants to Rif compared to WT? This would help to differentiate specific defects of RNAP mutations.

We determined the MIC and MBC of rifampicin adhering to established clinical guidelines by CLSI and EUCAST (PMID:18274517). While killing assays in the original version of the manuscript (Fig. 3h) are done in exponentially growing cells in LB broth with 1 hour of Rif treatment at 37C, MBC assays are done in Mueller-Hinton broth with a standardized inoculum for 18 hours at 37C. Again, we see a mild decrease in the MIC of hypersensitive mutants in MH broth (**new Extended Data Fig. 6c**). In addition, we determine the MBC of hypersensitive strains to be close to 2x their respective MIC (**new Extended Data Fig. 6d**). Rif remains bacteriostatic at the highest concentration tested in wild-type cells (80ug/mL) (**new Extended Data Fig. 6d**). As a result, the MBC/MIC ratio of hypersensitive strains is ≤ 4 while in wild-type cells it is >4 (**new Extended Data Fig. 6e**). These results demonstrate that Rif is bactericidal in hypersensitive strains across different experimental conditions.

3. Figure 5c: the legend seems to be not corresponding to the figure. It claims to be ???heatmap of rpoB mutant fitness and sensitivity or resistance to rifampicin, bicyclomycin, and 5-fluorouracil???, but the actual figure is a heatmap comparing the frequencies of mutations among different bacterial species.

We thank the reviewer for their careful reading of the manuscript. We apologize for this mistake and have corrected the legend.

4. It is mentioned that Rif is bactericidal to gram positive bacteria but bacteriostatic to *E. coli*, can this difference be attributed to different affinity of their RNAPs to Rif? Also, does Rif lethality in those organisms involve DNA damage? This can enhance generality of the model.

Previous work has shown that *M. tuberculosis* RNAP itself has no higher affinity for Rif *in vitro*; both *E. coli* and *M. tuberculosis* RNAP have an IC50 of between 2-10nM (PMID: 22570422, 18787125). In agreement, more recent work found no difference in Rif sensitivity when transplanting parts of the rifampicin binding site from *M. tuberculosis* to *E. coli* (PMID: 15793146). At present, very little is known about Rif lethality in gram-positives. Recent work has observed DNA damage to occur in *M. tuberculosis* after starving cells with PBS and then treating them with Rif, but it is uncertain whether this reflects physiological conditions (PMID: 28559332, 34818059). We believe that our study has the potential to advance our understanding of how Rif kills Gram positive bacteria, however further work will need to be done in those organisms.

REFEREE #3

The manuscript by Yang et al describes the generation of saturation mutagenesis mutants in the Rif binding site of RNAP and its mechanistic characterizations. The authors identified an alpha helix whose mutations was observed to have high sensitivity to Rif. Single residue mutants that increased the toxicity of Rif to a level beyond bacteriostatic to a bactericidal drug was also seen. Interestingly, the authors found mechanism of killing to be associated with DNA damage. Mutations that increased RNAP speed also showed increased sensitivity to other drugs and growth advantage in temperature shifts. The authors also performed some evolutionary analysis to look at the distributions of natural mutations in the Rif binding sites of RNAP in other bacteria.

Overall, this work appears to be done rigorously. The experiments and data presented are compelling and generally support the conclusions of the work. The work is pretty comprehensive in delineating the effects and consequences of these Rif binding site mutations. In terms of significance, there???'s some questions of whether the bactericidal effects of these mutations are clinically relevant given that they are generally not enriched under any natural situation. Recently another study identified compensatory mutations in RNAP under Rif resistance that increase RNAP rates (Doi: 10.1093/nar/gkac406), which is consistent with data shown in this manuscript. The work is a nice example of systematically generating mutants for more detailed mechanistic studies in understanding the multi-dimensional fitness landscape of antibiotic

resistance mutants. How these insights convert to better and more potent antimicrobial compounds remains to be seen.

We thank this reviewer for their helpful comments and agreement that our work is rigorous, compelling, and comprehensive. We have addressed all of the reviewer concerns as indicated below.

Some additional comments:

- Can the authors discuss why some of the STOP codon mutations are enriched or not highly selected against? (e.g. position 526 in Fig 3, positions 513, 517 in Extended Data Fig1, or positions 521, 569, 570, 572 in Extended Data Fig2)

All heatmaps with Rif, BCM, and 5FU (Fig. 3, Extended Data Fig. 2 and 3) are normalized for fitness such that they are displaying drug specific effects. In those comparisons, we would not expect stop codons to be selected against unless they provided additional drug sensitivity. Also, many of the stop codon positions that appear enriched after drug treatment are statistically insignificant and a result of variability in the low count of stop codons. We have provided additional heatmaps displaying each raw dataset prior to normalization between conditions (**new Extended Data Fig. 1**).

The stop codon mutants not depleted in the fitness screen (Extended Data Fig. 2) were more perplexing. A majority of positions are highly depleted, except for 513_ and 517_ as mentioned. We revisited the raw sequencing data at these positions and observed that rare single nucleotide substitutions sometimes lead to overcounting stop codons, however we intentionally designed MAGE oligos to maximize the mutational distance for each substitution. Originally, we directly translated each codon regardless of the designed MAGE oligo, because we underestimated the potential effects of background mutations. To address this, we modified the analysis pipeline to only look for the designed codon at each position. In addition, the fitness screen in the original submission was only done after a single 10 fold dilution of the mutant pool in liquid culture. This was done to reduce the chances of a few resistant mutations overwhelming the entire population in drug treated conditions.

To better observe stop codon depletion and provide a more comprehensive view on mutant viability, we repeated the MAGE fitness experiment with two additional timepoints in liquid after subsequent 100-fold dilutions, in addition to an experiment where we passaged the mutant pool on solid agar. In this extended fitness experiment we now see stop codons completely depleted as expected, and provide a more comprehensive catalog of viable Rif binding site mutants (**new Extended Data Fig 1, 2**).

- Do all fast RNAP mutants deplete nucleotides? The authors characterized a handful of mutants for characterization and the data is compelling overall, but I wonder if this is true for all observed mutants. Are there mutations that can be introduced in RNAP to slow it down to show that this is the major mechanism and not from other errant aspects of transcriptional changes?

This is an interesting point. While the current manuscript was primarily focused on characterizing drug sensitive mutants, our screening also identified several 5FU resistant mutants (Extended Data Fig. 4b). We noticed that one of the mutants identified, *rpoB* 513P, is a well

characterized Rif-resistant mutant known to be a slow polymerase (PMID: 10860976, 7526463). We reconstructed this mutant in MG1655 and verified that it is indeed more resistant to 5FU (**new Extended Data Fig. 16a**). This suggests that while a fast polymerase deplete nucleotides, a slow polymerase decreases the rate of thymidine consumption, providing more resistance to nucleotide analogues.

To provide further evidence that the slow RNAP 513P increases nucleotide reserves, we also measured its susceptibility to thymineless death by moving 513P into a *thyA* background. *thyA* mutants are unable to produce their own thymidine and must be supplemented with it to support growth. When thymidine is removed, cells begin a process known as thymineless death, resulting in cell lysis (PMID: 33318216). When compared to wild-type *thyA* cells that begin cell lysis after 1 hour of thymidine starvation, 513P is slower to deplete thymidine reserves and only begin losing viability after 2 hours (**new Extended Data Fig. 16b**). This also suggests that slow polymerases decrease the rate of nucleotide consumption.

In regard to errant transcriptional changes in fast RNAP mutants, we have also completed RNA-seq of the fast mutants 15Y and 27W and observe only small differences in genes involved in pyrimidine ribonucleotide biosynthesis and consumption. The only significant changes (log₂ fold change >2) are in the expression of *pyrD*, a gene involved in de novo pyrimidine ribonucleotide biosynthesis and in *upp*, a gene involved in pyrimidine ribonucleotide salvage (**new Extended Data Fig. 14a**).

To test if decreased levels of *pyrD* is responsible for differential 5FU sensitivity of fast mutants, we reconstructed fast mutants in a *pyrD* background and observed no rescue of 5FU sensitivity (**new Extended Data Fig. 14b**). *pyrD* deletion in wild-type cells also had little effect on 5FU sensitivity (**new Extended Data Fig. 14b**, Fig. 4g).

Upp downregulation in fast mutants is a more surprising finding since pyrimidine ribonucleotide salvage is also responsible for activation of 5FU. One would expect *upp* downregulation to increase resistance to 5FU, and this is indeed what we observe in wild-type cells and fast mutants lacking *upp* (**new Extended Data Fig. 14c**, Fig. 4g). However, *upp* fast mutants are still hypersensitive to 5FU when compared to *upp* wild-type cells (**new Extended Data Fig. 14c**). We conclude that transcriptional reprogramming is unlikely to be a primary mechanism for 5FU sensitivity in fast *rpoB* mutants.

The mechanism for differential regulation of *upp* and *pyrD* may be due to an additional, non-consequential attribute of fast mutants. The *carAB*, *pyrBI*, *upp-uraA*, and *codAB* operons are known to be regulated by a UTP-sensing mechanism of slippage and reiterative transcription while *pyrC* and *pyrD* are thought to respond to CTP by a different mechanism (see comprehensive review PMID: 18535147). The downregulation of *upp* and *pyrD* might suggest that fast mutants are more likely to slip and abort transcription of these genes. Alternatively, downregulation of *upp* might be a compensatory mechanism by cells to limit the conversion of imported uracil into uridine monophosphate, so that uracil can instead be used to produce deoxyuridine and then replenish dTMP through *deoA*, *tdk*, and *thyA*.

- Authors noted that the fast RNAP mutants are more fit in 30C vs WT where the screen was done but not at 37C. What conditions were the 5FU and BCM done? The authors showed data

for 25C and 42C for 5FU, but unclear it is unclear about 30C or 37C. Did the authors do the Rif screen at 37C, which is probably more clinically relevant?

We apologize for being unclear. All of the screening experiments were done at 30C because the EcNR2 strain necessary for MAGE recombineering is temperature sensitive. Raising the temperature in EcNR2 allows for recombineering to occur.

All other validation and mechanistic experiments (Fig. 3, 4) were done at 37C in the selected mutants reconstructed in wild-type MG1655 unless otherwise noted. All of the Rif MIC and killing experiments in addition to the BCM/5FU MIC, transcription speed, nucleotide measurement, and rescue experiments were done at 37C.

- Figure 4a, please show fit/correlation values with P-values.

We now provide these values in the figure legend as requested (**revised Fig. 4a**).

- The authors state in the discussion that "Rif kills cells through DNA damage and not transcription inhibition per se." The study shows that in hyper-sensitive Rif mutants, there is an additional component of DNA damage involved that yields more bactericidal effects. There should be more nuanced discussion of such points.

We have revised the discussion of this point as requested. To clarify, the established mechanism of Rif is thought to be due to transcription inhibition. How this would lead to cell death is unclear, and we show in this work that transcription inhibition is not the sole effector of the bactericidal effects of Rif.

In preparing this revision, we have greatly expanded on our model of DNA damage by Rif. First, we determined if DNA damage is a byproduct of reactive oxygen species (ROS). Seminal work from the Collins group described increased cellular respiration and buildup of toxic ROS as a common mechanism of cellular death induced by bactericidal antibiotics (PMID: 17803904). However, in that work, they saw no ROS increase in cells treated with bacteriostatic antibiotics, including Rif. Moreover, follow up work from the Collins lab studying the bactericidal effects of Rif in *S. aureus* found no increase in cellular respiration (PMID: 26100898), suggesting that the mechanism of killing by Rif is distinct from other bactericidal antibiotics. In agreement, we find that Rif killing does not entail increased ROS and cannot be rescued by antioxidants (**new Extended Data Fig. 9a-c**).

We then considered other possible sources for DNA damage. Prior work has shown that promoter-arrested RNAP presents an obstacle for the replisome, resulting in interference with replication fork progression or deleterious mutations at the promoter (PMID: 27362223, 16670199). These mutations at the promoter were dependent on *recA*, indicating that they are caused by recombination after replication fork collision (PMID: 27362223). This is consistent with our data that shows without *recA*, Rif becomes potently bactericidal in both wild-type and hypersensitive mutants (Fig. 3k).

We show in vitro that hypersensitive mutant RNAP remains bound to the promoter longer after Rif binding (Fig. 3f). We now also provide CHIP-seq data from wild-type and hypersensitive RNAP after recovery from Rif treatment and observe that more hypersensitive RNAP remains

bound to the promoter (**new Fig. 3g, new Extended Data Fig. 5a**). To investigate where DNA damage is occurring after Rif treatment, we designed a semi-long range qPCR assay (PMID: 19966269) to detect DNA breaks at the promoters and over the gene body of two highly expressed genes after Rif treatment (**new Extended Data Fig. 10a, b**). Interestingly, we observed a significant increase in DSB at gene promoters but not gene bodies (**new Fig. 3j**). From this data, we propose that replication-transcription conflicts at promoters contribute to DNA damage by Rif.

To further demonstrate that replication conflicts at promoters are involved in DNA damage by Rif, we introduced dCas9 targeting the *oriC* to hypersensitive strains to block replication prior to Rif treatment (**new Extended Data Fig. 10c**). We found that blocking replication significantly decreases the frequency of DSB breaks at promoters (**new Extended Data Fig. 10d, e**).

Next, we constructed hypersensitive mutants in cells containing a temperature-sensitive allele of the replication protein *dnaC2* (PC2) (PMID: 4925091, 20478253). We verified that shifting cells to the non-permissive temperature 42C inhibits replication (**new Extended Data Fig. 10f**). We then compared the bactericidal effects of Rif with and without replication. Strikingly, we observe that Rif is no longer bactericidal in hypersensitive mutants when replication is inhibited (**new Fig. 3l**). Rif remains bactericidal for hypersensitive mutants in the PC2 background at the permissive temperature of 30C (**new Extended Data Fig. 10g**).

In summary, we show that Rif induces DNA damage at promoters in a mechanism distinct from other bactericidal antibiotics, and the bactericidal effects and DNA damage by Rif are dependent on replication. This data, combined with previous work characterizing replisome conflicts with promoter-bound RNAP, provides evidence that Rif damages DNA through transcription-replication conflicts and not only through transcription inhibition.

In regard to the clinical relevance of our findings, we show that Rif potency can be increased in Gram-negatives, potentially expanding its treatment spectrum. Rif killing involves a unique mechanism of DNA damage, which implies that targeting DNA repair may increase Rif potency and can inform the design of combination antibiotic therapies. We note that even wild-type *E. coli* become susceptible to Rif killing when deficient in DNA repair (Fig. 3k). Improving our understanding of Rif is important because it is an essential antibiotic used in first line combination therapy for tuberculosis.

- In general, it's a little unclear whether the last analysis for Rif binding mutations in nature yielded any novel insights. Some of the results about conservation is not surprising. The observation of 24L enriched in species, which is a fast RNAP mutant in *E. coli*, is interesting. However, one cannot know for sure that such mutation also yields the fast RNAP phenotype in the rest of RNAP genetic background in those organisms so the results are speculative at best.

The evolutionary analysis was inspired by elegant work cited in our manuscript, using comparative genomics and MAGE-seq to characterize natural sigma 70 sequence diversity (PMID: 34433066). In that work, authors characterized the potential effects of sigma 70 orthologs on viability and gene expression. We thought that readers would have a similar interest in the occurrence of potentially functional mutations in *rpoB* orthologs.

We were also interested in whether the fast RNAP mutations we characterized appear in nature, because we observed that they can affect ideal growth conditions. Indeed, we did find such mutations in nature, guiding future work in those organisms characterizing the potential effects of the identified mutations. We agree that these mutants will need to be studied in their host organism to confirm their predicted effects. However, purification of RNAP from many of these organisms is not trivial and has not been done.

Nevertheless, the Rif binding site is highly conserved across evolution and there are many examples of Rif binding site mutations having comparable effects in different bacteria. For example, prior work from our lab has extensively characterized the clinically occurring Rif resistant mutation H526Y (PMID: 34697460). The major findings from that work include the observation that H526Y is a fast polymerase and served as the basis for the current manuscript. In that work, we generated the equivalent mutation in *M. tuberculosis* RNAP, H445Y, which also appears in clinical Rif resistance during tuberculosis infection. We found that this mutation in the *M. tuberculosis* RNAP genetic background also produces a fast RNAP. This supports our conjecture that modifications to this conserved and functional region in close proximity to the RNAP catalytic site would have similar effects on the speed of transcription.

Additionally, the effects of mutations conferring Rif resistance are also strikingly similar in different organisms. Mutations conferring rifampicin resistance in *M. tuberculosis* and *E. coli* are highly similar, and early structures of Rif resistant polymerases were solved in *Thermus aquaticus*, of the thermophilic Deinococcota phylum (PMID: 11290327). *Mycoplasma*, which carries the 526N mutation we identify in our evolutionary analysis, has already been shown to be resistant to rifampicin (PMID: 25118103). This provides further evidence that the Rif binding site has structural similarity in different organisms.

Reviewer Reports on the First Revision:

Referees' comments:

Referee #1 (Remarks to the Author):

The authors have done substantial work adequately addressing all of our concerns. One tiny little presentational suggestion: the order of the amino acids in Fig 1c should be the same as in Fig. 3a.

I am looking forward to seeing this paper in print.

Referee #2 (Remarks to the Author):

The revised manuscript by Yang et al provided many new experimental data addressing the key concerns regarding the antibiotic susceptibility mechanisms of the RNAP mutants. Overall, the revision addressed all my comments.

First, the revision provided multiple lines of support that Rif lethality of selected RNAP mutants is due to DNA damage rather than a symptom of cell death, by showing that Rif induces the SOS response using transcription-independent methods, as well as showing that Rif lethality corresponds to genetic manipulation of DNA repair systems but not reactive oxygen species. Second, the revision provided convincing data indicating that the Rif lethality is due to lethal double stranded breaks at the promoter region using SLR-qPCR, ChIP-seq, and replication block approach. Regarding the study of 5-FU susceptible RNAP mutants, the study ruled out the effect of transcription using RNAseq, and strengthened their nucleotide depletion model by showing that the mutants are also susceptible to other conditions of nucleotide depletion. The revision also cleared up the confusion arising from the methods of MIC and MBC measurements.

In terms of scientific content this is a solid and very interesting work worthy of publication in my opinion. A relatively minor limitation is how the information generated from these non-natural RNAP mutants can be translated to practical or physiological insights. Perhaps discussing their potential in drug designs, e.g. druggability of these specific RNAP regions, or how it may explain intrinsic Rif lethality in some bacterial species, can strengthen this aspect of the work.

Referee #3 (Remarks to the Author):

The revised manuscript by Yang is much improved with new data and analysis that greatly advance the overall work and its impact. All of the reviewer's prior concerns have been addressed. The reviewer appreciates the extensive experimental work done in the revision to further strength the

mechanistic insights and its novel contributions to the understanding of Rif-mediated cellular death.

Author Rebuttals to First Revision:

Referees' comments:

Referee #1 (Remarks to the Author):

The authors have done substantial work adequately addressing all of our concerns. One tiny little presentational suggestion: the order of the amino acids in Fig 1c should be the same as in Fig. 3a.

I am looking forward to seeing this paper in print.

We thank the reviewer for their work and supportive comments. We have modified Fig. 1c as requested.

Referee #2 (Remarks to the Author):

The revised manuscript by Yang et al provided many new experimental data addressing the key concerns regarding the antibiotic susceptibility mechanisms of the RNAP mutants. Overall, the revision addressed all my comments.

First, the revision provided multiple lines of support that Rif lethality of selected RNAP mutants is due to DNA damage rather than a symptom of cell death, by showing that Rif induces the SOS response using transcription-independent methods, as well as showing that Rif lethality corresponds to genetic manipulation of DNA repair systems but not reactive oxygen species. Second, the revision provided convincing data indicating that the Rif lethality is due to lethal double stranded breaks at the promoter region using SLR-qPCR, ChIP-seq, and replication block approach. Regarding the study of 5-FU susceptible RNAP mutants, the study ruled out the effect of transcription using RNAseq, and strengthened their nucleotide depletion model by showing that the mutants are also susceptible to other conditions of nucleotide depletion. The revision also cleared up the confusion arising from the methods of MIC and MBC measurements.

In terms of scientific content this is a solid and very interesting work worthy of publication in my opinion. A relatively minor limitation is how the information generated from these non-natural RNAP mutants can be translated to practical or physiological insights. Perhaps discussing their potential in drug designs, e.g. druggability of these specific RNAP regions, or how it may explain intrinsic Rif lethality in some bacterial species, can strengthen this aspect of the work.

We thank the reviewer for their work and supportive comments. We have expanded in the discussion on how our results can potentially improve drug design and help understand intrinsic Rif lethality.

Referee #3 (Remarks to the Author):

The revised manuscript by Yang is much improved with new data and analysis that greatly advance the overall work and its impact. All of the reviewer's prior concerns have been addressed. The reviewer appreciates the extensive experimental work done in the revision to further strength the mechanistic insights and its novel contributions to the understanding of Rif-mediated cellular death.

We thank the reviewer for their work and supportive comments.